# Structural and functional insights into the modulation of T cell costimulation by monkeypox virus protein M2

Shangyu Yang[1,5], Yong Wang[2,3,5], Feiyang Yu[1,5], Rao Cheng[1], Yiwei Zhang[1], Dan Zhou[1], Xuanxiu Ren[1], Zengqin Deng ●[2,4] ✉ & Haiyan Zhao ●[1] ✉

The rapid spread of monkeypox in multiple countries has resulted in a global public health threat and has caused international concerns since May 2022. Poxvirus encoded M2 protein is a member of the poxvirus immune evasion family and plays roles in host immunomodulation via the regulation of innate immune response mediated by the NF-κB pathway and adaptive immune response mediated by B7 ligands. However, the interaction of monkeypox virus (MPXV) M2 with B7 ligands and structural insight into poxviral M2 function have remained elusive. Here we reveal that MPXV M2, co-existing as a hexamer and a heptamer, recognizes human B7.1 and B7.2 (hB7.1/2) with high avidities. The binding of oligomeric MPXV M2 interrupts the interactions of hB7.1/2 with CD28 and CTLA4 and subverts T cell activation mediated by B7.1/2 costimulatory signals. Cryo-EM structures of M2 in complex with hB7.1/2 show that M2 binds to the shallow concave face of hB7.1/2 and displays sterically competition with CD28 and CTLA4 for the binding to hB7.1/2. Our findings provide structural mechanisms of poxviral M2 function and immune evasion deployed by poxviruses.

Monkeypox virus (MPXV), an *Orthopoxviridae* family member, was first identified in 1958 from laboratory monkeys and is believed to be mainly maintained in rodents and small animals with sporadic human infection. Since the first human case was reported in Africa in 1970, it primarily circulated in West and Central Africa[1,2]. Most MPXV-infected patients develop classical skin lesions and maculopapular rash with common symptoms consisting of fever, headache, lethargy, and myalgia. Monkeypox is generally considered a self-limited disease; however, severe complications have been reported, including encephalitis, permanent vision loss, and septicemia, especially in immunocompromised individuals[2]. Recently, more than 100 countries outside of Africa have reported 84,075 MPXV-infected human cases with 76 deaths until 4 Jan 2023, posing a great challenge to public health.

The B7 family proteins B7.1 (CD80) and B7.2 (CD86) are two widely studied glycoproteins and play critical roles in host immunity against contagious pathogens and cancers[3,4]. These two molecules are primarily expressed by immune cells, including dendritic cells (DCs) and B cells, and can interact with the costimulatory receptor CD28 for T cells activation, maturation, and function. They also interact with the inhibitory receptor cytotoxic T lymphocyte-associated antigen 4 (CTLA4) on T cells to keep immune cells under check for optimal immune response and homeostasis. Several infection pathogens and tumors have been reported to target these B7.1/2-CD28/CTLA4 pathways to build an immunosuppressive microenvironment for better replication or survival[5–10].

Orthopoxviruses are double-stranded DNA viruses, including the causative agent of smallpox (variola virus, VARV), cowpox virus

[1]State Key Laboratory of Virology, College of Life Sciences, Wuhan University, Wuhan, Hubei, China. [2]Center for Antiviral Research, Wuhan Institute of Virology, Chinese Academy of Sciences, Wuhan, Hubei, China. [3]University of Chinese Academy of Sciences, Beijing, China. [4]Hubei Jiangxia Laboratory, Wuhan, Hubei, China. [5]These authors contributed equally: Shangyu Yang, Yong Wang, Feiyang Yu. ✉e-mail: dengzengqin@wh.iov.cn; haiyzhao@whu.edu.cn

(CPXV), and vaccinia virus (VACV)[11]. Orthopoxviruses encode up to 200 viral proteins, and a large number of these proteins are involved in virus-host interaction to facilitate viral replication[12,13]. One of these proteins, M2, is expressed in the early stage of viral infection and secreted by infected cells, presenting multiple functions in viral infection. VACV M2 protein decreases the phosphorylation of ERK2 and inhibits NF-κB activation[14]. In addition, M2 is indicated to play roles in virus uncoating and DNA replication based on its compensating effect of 68k-ank protein (68-kDa ankyrin repeat/F-box protein)[15,16]. More recently, two independent studies showed that some poxviral M2 proteins could recognize human/mouse B7.1 and B7.2, leading to the blockade of T cell activation and proliferation[17,18]. Enzyme-linked immunosorbent assay (ELISA) and cell-based flow cytometry assay also indicated that the binding of M2 to B7.1 or B7.2 interferes with the interactions between B7.1/2 and soluble CD28 or CTLA4, whereas the interaction between B7.1 and PD-L1 (programmed death ligand 1) is enhanced after M2 engagement[17,18], which is a negative costimulatory molecular and involved in negative regulation of the T cell immune responses[19,20]. However, whether MPXV M2 has a similar function as reported for VACV and CPXV M2 is not clear, and the molecular basis underlying how poxviral M2 engages human B7.1/2 and sabotages T cell activation and proliferation remains elusive.

Here, we show that MPXV M2 can interact with human B7.1 and B7.2 and block the binding of CD28 and CTLA4 to the human B7.1/2 ligands. Importantly, oligomeric M2 can inhibit T cell activation mediated by B7.1/2 costimulatory pathways, whereas this inhibition effect is dramatically reduced using monomeric M2. We also determine the cryo-EM structures of MPXV M2 complexed with human B7.1 and B7.2 at resolutions between 2.7 and 3.1 Å. The key contacts in the complexes on the MPXV M2 protein are strictly conserved among different orthopoxviruses, suggesting a commonly used immune evasion mechanism. Overall, this work provides the first structural glimpse of M2 protein from orthopoxviruses and defines how MPXV M2 interacts with human B7.1/2 and impacts T cell function, promoting our understanding of poxvirus immune evasion.

## Results

### MPXV M2 recognizes B7.1 and B7.2 of B7 family

To test whether MPXV M2 can recognize human B7.1 (hB7.1) and B7.2 (hB7.2), we first expressed and purified biotinylated MPXV Avi-M2 protein (Bio-M2) from the cell culture supernatant of Expi293 cells (Supplementary Fig. 1). Meanwhile, we transiently transfected Expi293 cells with hB7.1 and hB7.2 and performed flow cytometry to assess the binding of MPXV M2 to hB7.1 and hB7.2 on the Expi293 cell surface. A clear fluorescence shift was observed at the concentration of 0.32 nM M2 to the cells expressing hB7.1 or hB7.2, and the fluorescence signals on the cell surface were gradually increased along with increased M2 concentrations. In addition, MPXV M2 can also bind to cells expressing mouse B7.1 (mB7.1) and B7.2 (mB7.2). These results suggest that MPXV M2 can dose-dependently bind to human B7.1 and B7.2 as well as mouse B7.1 and B7.2 (Fig. 1a). We also found that VACV and CPXV M2 proteins can interact with human and mouse B7.1/2 under our experimental conditions, which is consistent with previous studies[17,18] (Fig. 1b, c).

We next quantitatively assessed the binding capacity of MPXV M2 to hB7.1 and hB7.2 via biomolecular layer interferometry (BLI) (Fig. 1d). Purified Bio-M2 protein was immobilized onto streptavidin biosensors, and the sensors were dipped into serial dilutions of hB7.1 or hB7.2. The BLI analyses showed that MPXV M2 binds to hB7.1 and hB7.2 efficiently, with a kinetically derived binding affinity ($K_D$) of 95.3 nM and dissociation half-life time (t1/2) of 102.5 s for hB7.1, and a $K_D$ of 388 nM and t1/2 of 20.7 s for hB7.2, respectively. Generally, the costimulatory molecules B7.1/2 are expressed on the surface of antigen-presenting cells and interact with CD28/CTLA4 to regulate T-cell response, and the M2 protein is believed to be secreted by virally infected cells. To

measure the interactions of M2 with hB7.1/2 in a more physiological context, purified hB7.1/2-Fc proteins were immobilized onto protein A biosensors, and the sensors were dipped into serial dilutions of MPXV M2 (Fig. 1e). The binding signals were recorded, and the experimental curves were fitted using a 1:1 binding model to calculate the average binding affinity. Surprisingly, the binding ability and half-life of M2 with hB7.1/2 significantly increased in this setting, leading to a $K_D$ of 0.15 nM and t1/2 of 397 min for hB7.1-M2 complex and a $K_D$ of 0.05 nM and t1/2 of 1180 min for hB7.2-M2 complex, respectively. These results suggest that the interactions of M2 with hB7.1/2 are much tighter than the binding of CD28/CTLA4 to hB7.1/2, which has been reported in the range of 0.2–20 μM. MPXV M2 can interact with both human and mouse B7.1/2; nonetheless, it binds to mouse B7.2 with relatively weaker affinity than human B7.1/2 (Supplementary Fig. 2).

In addition to B7.1 and B7.2, other proteins in the B7 family are also involved in T cell immune responses by interacting with the CD28 family receptors. To further verify whether MPXV M2 binds to other B7 ligands on the cell surface, we transiently transfected Expi293 cells with several other human B7 family members, including hPD-L1, hPD-L2, hB7-H3, hB7-H4, hB7-H5, hB7-H6, hB7-H7, and hILDR2 for the binding investigation via flow cytometry. Among tested B7 family ligands, only hB7.1- and hB7.2-transfected Expi293 cells can be recognized by MPXV M2, indicating that MPXV M2 can specifically bind to B7.1 and B7.2 on the cell surface (Supplementary Fig. 3).

### Cryo-EM structures of the M2-hB7.1 and M2-hB7.2 complexes

To understand the structural basis of the interaction between M2 and human B7.1/2, we sought to determine the structures of M2 alone and M2-hB7.1/2 complexes using single-particle cryo-electron microscopy (cryo-EM). Due to severe preferred orientation issues, we could not resolve the cryo-EM structure of M2 alone. We then focused on the structural determination of M2-hB7.1/2 complexes. M2 was mixed with molar excess hB7 protein and then subjected to cryo-EM analysis. Interestingly, two oligomeric states, hexamer and heptamer, were revealed in both cryo-EM samples (Supplementary Figs. 4, 5), which were also observed in 2D class averages of M2 alone (Supplementary Fig. 8b). We could not obtain a three-dimensional reconstruction of the M2-hB7.1 heptamer complex, partially due to the few cryo-EM particles of this complex on the cryo-EM grids. We finally determined three complex structures, M2-hB7.1 hexamer, M2-hB7.2 hexamer, and M2-hB7.2 heptamer, at an overall resolution of 3.04, 2.7, and 3.12 Å, respectively (Supplementary Figs. 4, 5 and Table 1). The quality of the cryo-EM density maps is sufficient for unambiguous residue assignment guided by Alphafold2 predicted M2 model and crystal structures of human B7.1 and B7.2. The densities of the C-terminal domain of hB7.1/2 were weak; thus, the corresponding parts were excluded from the refined models. The resulting structures reveal that M2 forms an oligomeric ring structure with B7 proteins binding to the outer face of the M2 ring in a 1:1 stoichiometry (Fig. 2a, b and Supplementary Fig. 6a). Despite the different oligomeric conformations, the overall structure of each M2 subunit and inter-subunit interactions from hexamer and heptamer structures are virtually identical (Supplementary Fig. 6). The root-mean-square deviation (r.m.s.d) is ~0.4 Å for all Cα atoms of M2 subunits of two oligomeric structures. The subtle differences arise from connecting loops that appear more dynamic owing to fewer protein-protein interactions.

### Architecture of the MPXV M2 oligomer

Based on its predicted structure, M2 has been classified into the poxvirus immune evasion (PIE) domain superfamily, which features a β-sandwich core domain and is involved in the manipulation of host responses to viral infection[21]. In our cryo-EM structures, the M2 subunit forms a two-layer β-sandwich fold consisting of two parallel β-sheets, which are stabilized by an intramolecular disulfide bond and the connecting loops (Fig. 2c). The front β-sheet contains five antiparallel

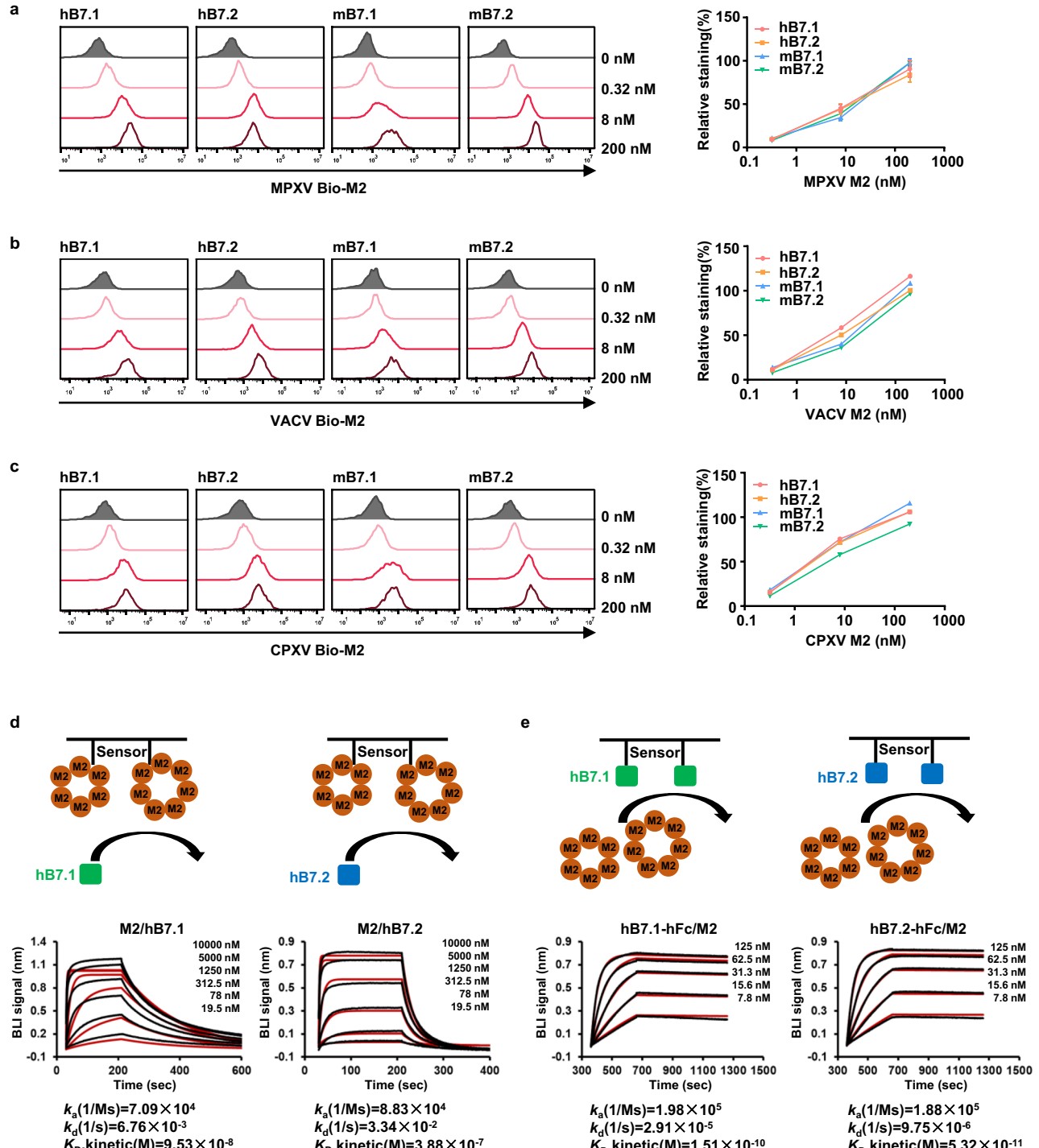

**Fig. 1 | M2 specifically binds B7.1 and B7.2. a–c** The interactions of MPXV M2 with B7 ligands were evaluated by flow cytometry. Expi293 cells transiently transfected with human and mouse B7.1 or B7.2 (Expi293-hB7.1/2 or Expi293-mB7.1/2) were pre-incubated with the indicated concentrations of Bio-M2, and the cells were then stained with 0.2 μg/mL secondary streptavidin-APC antibody. The cells expressing corresponding B7 ligands stained with secondary antibody alone were used as a negative control (black). Representative flow cytometric curves were shown for MPXV Bio-M2 (**a**), VACV Bio-M2 (**b**), and CPXV Bio-M2 (**c**) (left). The average mean fluorescence intensity (MFI) from three (**a**) or two (**b**, **c**) independent experiments performed in duplicates is shown, and data in right panel of (**a–c**) are shown as

mean or mean ± SEM (right). Gating strategy is shown in Supplementary Fig. 11a. **d**, **e** Quantitative analysis of the binding affinity of MPXV M2 to hB7.1/2 by BLI. The binding curves were obtained by immersing M2 immobilized sensors into serially diluted hB7.1/2 (**d**), or by immersing hB7.1/2 immobilized sensors into different concentrations of M2 (**e**). The sensors without M2 or hB7.1 loading were used in parallel to define the background. The BLI traces from one of three independent experiments are shown. The kinetic values were obtained by simultaneously fitting the association and dissociation responses to a 1:1 Langmuir binding model ($K_D$, kinetic). Source data are provided as a Source data file.

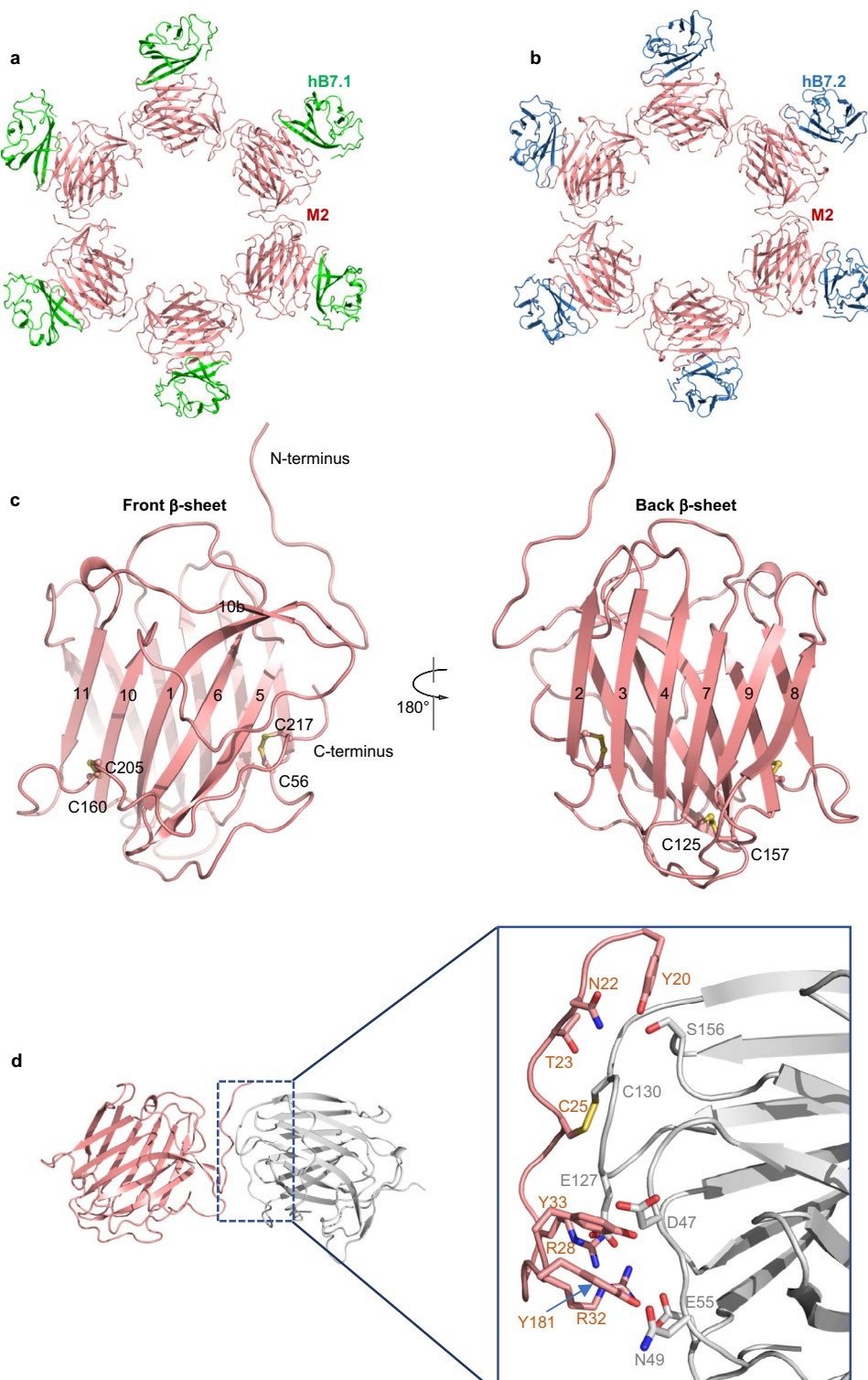

**Fig. 2 | Cryo-EM structures of the M2-hB7.1 and M2-hB7.2 complexes. a, b** The overall structures of M2-hB7.1 hexamer (**a**) and M2-hB7.2 hexamer (**b**). M2, hB7.1 and hB7.2 are colored salmon, green and skyblue, respectively. **c** Structure of a single M2 subunit. **d** M2 subunit interface of M2-hB7.1 hexamer. Disulfide bonds are indicated with yellow sticks.

strands β1, β5, β6, β10, and β11, whereas the back β-sheet consists of six strands, which can be further divided into two antiparallel β-sheets (β2-β3-β4-β7 and β8-β9 sheets). The N-terminus of M2 is an extended loop responsible for subunit interaction. A long loop connecting β10 and β11 decorates the surface of the front β-sheet with a short β strand (β10b) paralleling the β1 strand. The C-terminal loop following β11 runs across the bottom of the front β-sheet and then wraps the side surface

of the β-sandwich fold. The conformation of this long C-terminal loop is fixed by two intramolecular disulfide linkages (Fig. 2c). Compared to the available structures of the PIE domain superfamily, the overall fold of the β-sheet core domain is highly similar, but M2 has the most structural similarity with the SECRET (smallpox virus-encoded chemokine receptor) domain of ectromelia virus (Supplementary Fig. 10a)[22].

The inter-subunit interface buries ~750 Å$^2$ of molecular surface per subunit. Residues D47, N49, and E55 from the β1-β2 loop, E127 from the β7-β8 loop and S156 from the β9-β10 loop along with the N-terminal residues of neighboring M2 molecule interdigitate through a network of both hydrogen bonding and electrostatic interactions. The M2 inter-subunit interactions are further strengthened by one intermolecular disulfide bond formed by C25 and C130 from adjacent subunits (Fig. 2d and Supplementary Table 2). The residues involved in the subunit-subunit interface are highly conserved across orthopoxvirus M2 (Supplementary Fig. 10b). The existence of intermolecular disulfide linkage and different oligomeric conformations were further confirmed by SDS-PAGE analysis. M2 purified by size exclusion chromatography was loaded to SDS-PAGE analysis in the absence or presence of reduced reagent. The expressed monomeric MPXV M2 harboring four predicted N-linked glycosylation sites, which is consistent with our resolved cryo-EM densities (Supplementary Fig. 4e), is expected to be 36.2 kDa (one protein with four N-glycans). Two distinct bands with molecular weights between 150 and 250 kDa were observed in the absence of reduced reagent 2-Mercaptoethanol (BME), whereas only one single band with a molecular weight of about 35 kDa appeared in the presence of BME (Supplementary Fig. 1a). These results suggest that M2 forms multiple oligomers via disulfide bonds in solution. Interestingly, like MPXV M2, two bands of M2 from the other two poxviruses (VACV and CPXV) representing oligomeric states were reduced to a monomeric state with the introduction of BME (Supplementary Fig. 1c, d), suggesting that poxvirus M2 proteins have a conserved oligomer organization via an intermolecular disulfide bond.

## Oligomeric assembly of recombinant M2 and viral secreted M2

Cryo-EM structures show that the disulfide bond formed by C25 from one M2 molecule and C130 from adjacent M2 contributes significantly to the oligomerization of M2. To further confirm the oligomeric assembly and investigate the multivalent effects of M2, we substituted C25 and C130 with Ser and purified this M2 variant (M2-SS). M2-SS was eluted sharply delayed compared with wild-type M2 protein on size-exclusion chromatography (SEC), and only one band was observed on SDS-PAGE gel in both reduced and non-reduced conditions (Supplementary Figs. 7a, 8a). The size exclusion chromatography coupled with multi-angle light scattering (SEC-MALS) determined the protein component molecular weight of M2-SS is 25.8 ± 0.452% kDa (Supplementary Fig. 8d), which is close to the expected monomeric molecular weight (25.3 kDa), suggesting the recombinant M2-SS exists as a monomer in solution. Meanwhile, the average protein component molecular weight of wild-type M2 is 154.9 kDa ± 0.557%, corresponding to ~6.12× M2 monomer (Supplementary Fig. 8). The BLI analysis showed that monomeric M2-SS can still bind to hB7.1/2, with a 100-fold lower binding affinity (827 nM for hB7.1 and 1080 nM for hB7.2) and relatively fast dissociation rate (28.4 sec for hB7.1 and 20.6 sec for hB7.2) compared with oligomeric M2 (Supplementary Fig. 7b, c). These data corroborate our structural finding that MPXV M2 forms homo-oligomers.

To delineate the M2 assembly in the face of viral infection, culture supernatant from vaccinia-infected cells was probed with hB7.1-Fc and hB7.2-Fc. We here used vaccinia virus (vTF7-3[23]) as an indicator for the oligomeric feature of MPXV M2 mainly due to the biosafety concern of working with MPXV and the high conservations of M2 among orthopoxviruses (amino acid sequence identity >92%). The western blot analysis shows that three high molecular weight bands were detected by hB7.1/2 in culture medium from vaccinia virus-infected BHK21 cells under the non-reduced condition, and the primary two bands run at comparable positions as recombinant VACV and MPXV M2 proteins (Supplementary Fig. 9), indicating that secreted M2 during viral infection also forms high molecular weight homo-oligomers. An inapparent, even larger size band was also observed on the western blot, and the assembly of this species from the actual virions awaits

further characterization. Consistent with our results, secreted M2 from vaccinia virus (Copenhagen)-infected CEFs was also visualized as large oligomers by hB7.1/2, and M2 hasn't been detected in the supernatant of vaccinia virus (MVA) infected cells, which is a highly attenuated and modified vaccinia virus Ankara (MVA) with the M2 gene deletion[18].

## Interactions between MPXV M2 and hB7.1/2

Structural analysis reveals that M2 interacts with hB7.1 and hB7.2 similarly (Fig. 3). The interfaces of the two complex structures have very similar atomic contacts and bury a total accessible surface area of 1760 and 1617 Å$^2$, respectively. The structures of the hB7.1/2 monomer in the complex are highly similar to that observed in CTLA4-hB7.1/2 complex structures (Fig. 4a, c), displaying r.m.s.d. of 0.68 and 0.65 Å, respectively. M2 binds to hB7.1/2 side-by-side via β-strand complementation mediated by the C-terminal half of the M2 β11 and the N-terminal part of the hB7.1/2 G strand. The long C-terminal loop of M2 sits on top of the shallow concave surface, formed by the natural twist of strands G, F, C, and C′ of hB7.1/2 (Fig. 3a, b). Extensive residues from the C-terminal loop, β8-β9 loop, and β10-β11 loop, and to a lesser extent, the β8, β9, β10, and β11 strands form a network of both van der Waals and hydrogen bonding interactions with residues from strands G, F, C, and C′ of hB7.1/2. Specifically, residues E122, K123, F126, K127, R128, and E133 from hB7.1, and residues M120, R122, Q125 as well as N127 from hB7.2 form hydrogen bonds with corresponding residues of M2 protein (Fig. 3c–f; Supplementary Table 3).

## MPXV M2 disrupts the interactions of CD28 and CTLA4 with B7.1 and B7.2

To better understand the mechanism of action of M2, we compared the M2's binding footprints on hB7.1 and hB7.2 with the footprints of CD28 family members. To our knowledge, no structural information on CD28 complexed with B7 ligands is available; however, accumulated studies suggest that CD28 shares a common protein fold with CTLA4 and has significantly overlapped interacting residues on B7 ligands as CTLA4[24,25]. We aligned the M2-hB7.1 and M2-hB7.2 structures with the existing CTLA4-hB7.1 (PDB: 1I8L)[26] and CTLA4-hB7.2 (PDB: 1I85)[27] structures, respectively. We found that M2 and CTLA4 engage to the surface of GFCC′ strands of hB7.1/2 similarly. The interfaces between M2 and the B7 proteins overlay the contact surfaces of CTLA4 and B7 ligands (Fig. 4). There is a total of 12 residues on GFCC′ strands as well as CC′- and FG-loops of hB7.1 reside in the CTLA4-hB7.1 interface; however, 8 of them are also contacted by M2 (Fig. 4b, e). For hB7.2, CTLA4 makes 14 contacts with hB7.2 and M2 makes 15 contacts with hB7.2, with 8 of these being shared residues (Fig. 4d, e). CD28 and CTLA4 are believed to recognize the overlapped sites on B7 ligands, and our structural comparisons suggest that binding of M2 to the hB7.1/2 could sterically block the interactions of B7 ligands with CD28 family receptors.

To evaluate the blockade activity experimentally, we used lentivirus transduced 293T cells which can express hB7.1 and hB7.2 on the cell surface to investigate the competition activity between M2 and human CD28/CTLA4 (hCD28/hCTLA4) for hB7.1/2 binding. As expected, when we incubated hB7.1/2-expressing 293T cells with recombinant hCD28-Fc or hCTLA4-Fc, a dramatical fluorescence shift was observed (Fig. 5a–d), indicating that these two proteins could recognize cells containing hB7.1/2. However, the binding signal of hCD28 or hCTLA4 to hB7.1/2 was significantly reduced in the presence of M2, indicating that M2 affected the binding of hCD28 and hCTLA4 to hB7.1/2 on the cell surface (Fig. 5a–d).

We further verified this with a competitive BLI assay. The protein A biosensors were used to load hB7.1-hFc or hB7.2-hFc. Following this, the binding capacity of hCD28 and hCTLA4 to hB7.1/2 was determined in the presence or absence of M2 (Fig. 5e–h). Indeed, hCD28 and hCTLA4 can bind to immobilized hB7.1/2 with great signals (BLI signals: -0.52 for hB7.1/CD28, -1.20 for hB7.1/CTLA4, -0.74 for hB7.12/CD28, and -1.33 for hB7.12/CTLA4), while after M2 bound to immobilized

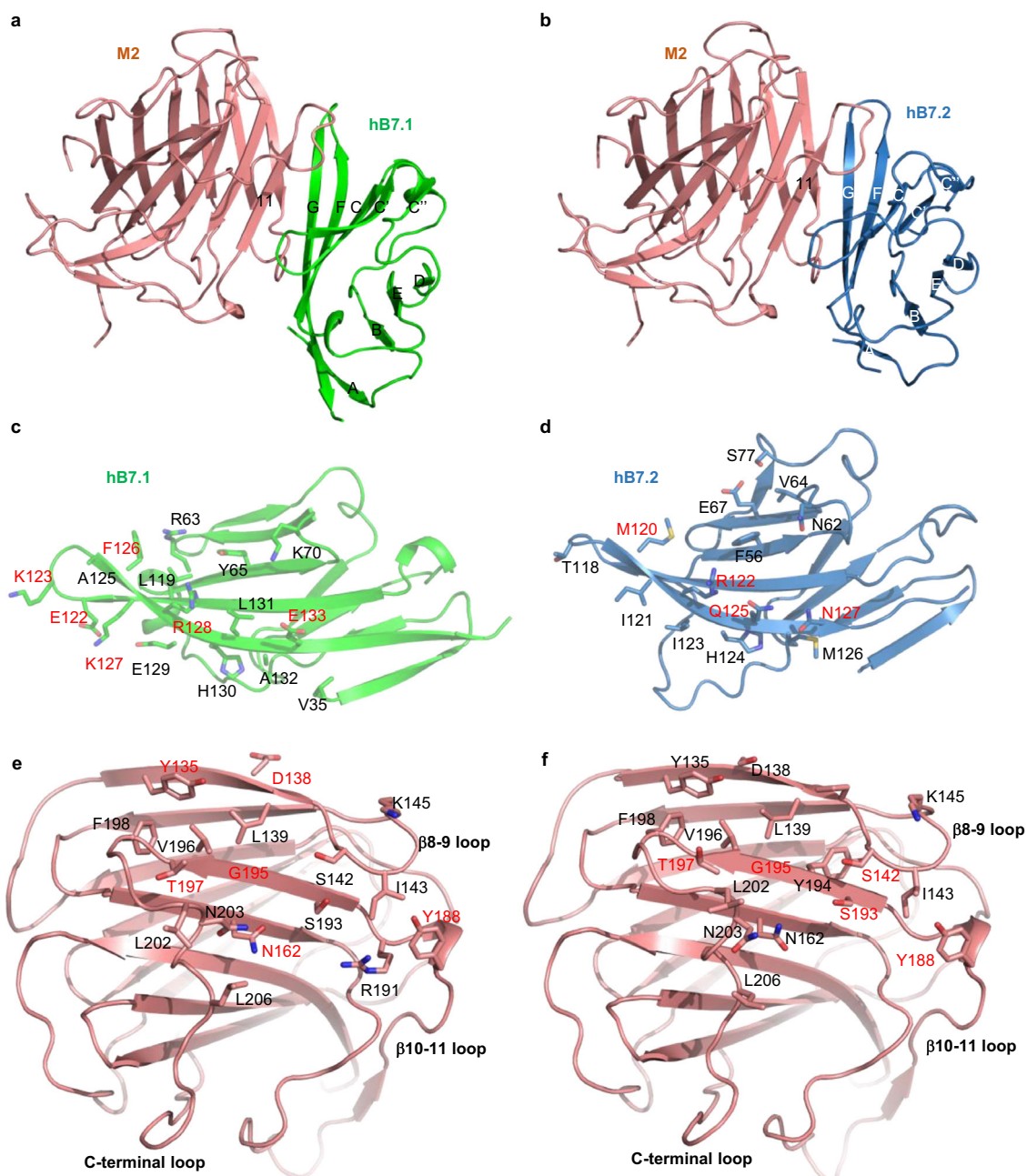

**Fig. 3 | Interactions between M2 and hB7.1/2. a, b** One subunit of M2-hB7.1 and M2-hB7.2 complexes. **c, d** The structures of hB7.1 of M2-hB7.1 hexamer (**c**) and hB7.2 of M2-hB7.2 hexamer (**d**). **e, f** The structures of M2 of M2-hB7.1 hexamer (**e**) and M2 of M2-hB7.2 hexamer (**f**). Residues involved in the interactions between M2 and hB7.1/2 are shown as sticks, and interacting residues form hydrogen bonds are labeled in red. The interactions were determined using LigPlot+ with a cutoff distance of 3.9 Å (Supplementary Table 3).

hB7.1 or hB7.2 proteins, the binding of hCD28 or hCTLA4 to hB7.1/2 were decreased sharply. These results strongly support our structural analysis and illustrate how M2 may modulate T cell immune response by partially hijacking the B7-mediated signal pathways.

## MPXV M2 impairs T cell activation mediated by B7 costimulation

B7-mediated signaling is required for T cell immunity, we then investigated the biological effects of M2 on hB7.1/2 costimulatory functions via an ex vivo T cell activation assay. Purified T cells from peripheral blood mononuclear cells (PBMC) of healthy humans were co-incubated with hB7.1/2-expressing cells which provide costimulatory signal 2, and anti-human CD3 antibody was used for the activation of signal 1 for this

experiment. PMA/Ionomycin can activate T cells independent of the costimulatory signal provided by B7.1/2 and was included to serve as the positive control. IL-2 cytokine production was observed when co-culture T cells with 293T-hB7.1/2 cells or stimulating T cells with PMA/Ionomycin, indicating T cells were activated in both conditions (Fig. 6a). As expected, IL-2 production was significantly reduced in the presence of MPXV M2 protein activated with anti-human CD3 antibody plus hB7.1/2. However, PMA/Ionomycin-derived T cell activation could not be antagonized in the presence of M2 (Fig. 6a), suggesting MPXV M2 explicitly inhibits T cell activation mediated by B7 costimulatory molecules. This is in line with previous studies that mouse B7-mediated T cell activation was sabotaged by CPXV M2 protein, and culture supernatant from M2-deficient CPXV-infected cells cannot inhibit IL-2

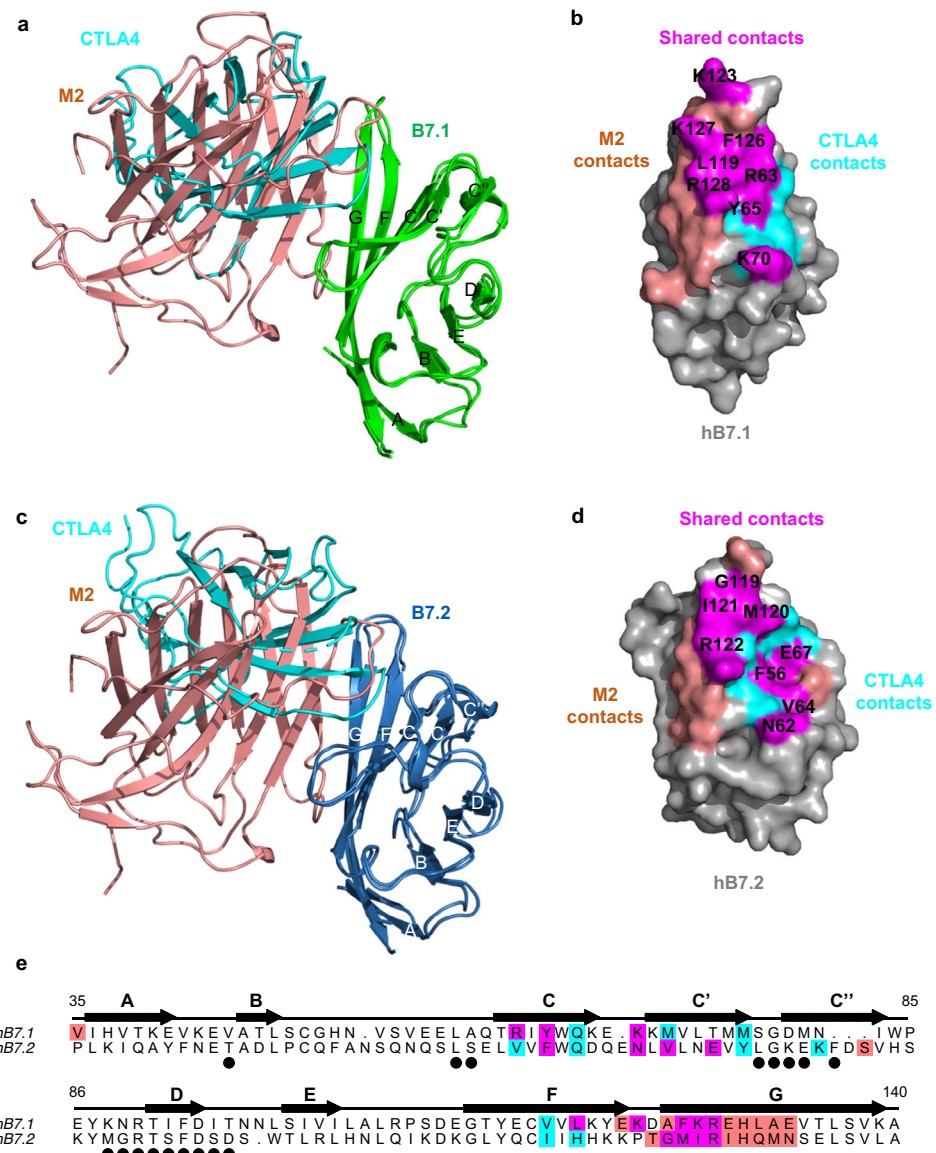

**Fig. 4 | Footprints comparison of MPXV M2 and CTLA4 binding to human B7.1/2. a** Superimposition of M2-hB7.1 complex and CTLA4-hB7.1 complex (PDB:1I8L) by aligning hB7.1. **c** Superimposition of M2-hB7.2 complex and CTLA4-hB7.2 complex (PDB: 1I85) by aligning hB7.2. **b**, **d** The surfaces of hB7.1 and hB7.2 are shown with the binding sites colored: salmon for the M2 binding footprints, cyan for the CTLA4 binding footprints, and magenta for the shared contacts by M2 and hB7.1/2. **e** Sequence alignment of hB7.1 and hB7.2. The binding residues of M2 and CTLA4 in hB7.1 and hB7.2 are colored as in (**c**) and (**d**). The residues involved in CTLA4 dimer interface are marked with black circles. The interactions were determined using LigPlot+ with a cutoff of 3.9 Å.

production compared with that from CPXV-WT infected cells. Of note, the ability of IL-2 inhibition by monomeric MPXV M2 was dramatically weakened than the oligomeric M2, supporting the notion that the high-avidity binding property of oligomeric M2 to hB7.1/2 is important for viral evasion of T cell response (Fig. 6b).

## Discussion

M2 protein and its orthologues are broadly expressed in most ortho-poxviruses, and it has been reported able to modulate the innate and adaptive immune response in the context of orthopoxvirus infection; however, the roles of M2 among different orthopoxviruses and mechanisms of action remain unclear. In this study, we performed a structural analysis of MPXV M2 complexed with two extensively studied costimulatory molecules, B7.1 and B7.2, providing new insights into poxvirus immune evasion.

Recombinant CPXV M2 can affect B7-mediated T cell activation and cytokine production, and M2-deleted CPXV induced a stronger

cellular immune response in mice than WT CPXV[17]. Another independent study shows that a vaccinia Copenhagen strain with the M2 gene deletion has little effect on VACV replication and tumor inhibition, even the culture supernatant from M2-deleted VACV infected cells does not have any B7-interfering activities[18]. Modified vaccinia virus Ankara (MVA), a highly attenuated VACV, is widely used as viral vector in many clinical trials against infectious diseases and cancers[28]. MVA has lost a lot of immune modulation genes during serial passages, including M2, and could induce robust T-cell immune responses against antigens[29,30]. MPXV with a genomic region deletion containing M2 is strongly attenuated[31]. The MPXV genomic surveillance is still going on, and some genome analyses indicate that gene gain and loss might contribute to the adaption of MPXV to humans for the ongoing outbreak[32–34], and one 2022 USA MPXV strain shows significant genomic rearrangement, including an M2 gene duplication[35]. Even though it's difficult to predict each gene's contribution to viral virulence and immunomodulation, MPXV

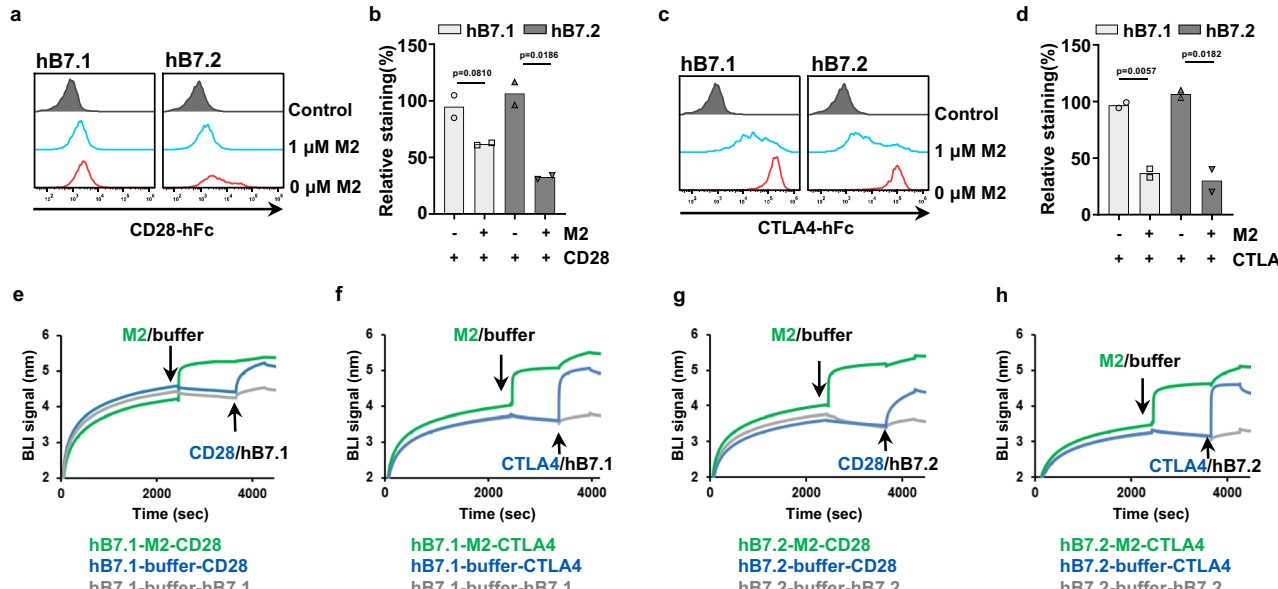

**Fig. 5 | MPXV M2 affects the interactions of B7.1/2 with CD28 and CTLA4. a, b** A flow cytometry analysis of the role of M2 on the binding of CD28 to hB7.1/2. **c, d** A flow cytometry analysis of the role of M2 on the binding of CTLA4 to hB7.1/2. hB7.1-transduced (293T-hB7.1) and hB7.2-transduced (293T-hB7.2) cells were first stained with human CD28-hFc or CTLA4-hFc proteins in the presence or absence of the MPXV M2, then the binding capacity of CD28-hFc and CTLA4-hFc to the 293T-hB7.1/2 was visualized by secondary antibodies (anti-human IgG/488) via flow cytometry. Representative flow cytometric plots of two independent experiments performed in duplicate (**a, c**). The data averaged from two independent experiments are shown as mean (**b, d**). 293T-hB7.1/2 cells stained with secondary antibody

alone were used as a negative control (black). Gating strategy is shown in Supplementary Fig. 11b. Data were analyzed by two-tailed unpaired Student's t test. *P < 0.05; **P < 0.01. **e–h** CD28/CTLA4 and M2 competition for the binding to hB7.1 (**e, f**) and hB7.2 (**g, h**) were tested by BLI. hB7.1/2-hFc were first loaded on the protein A biosensors, then the biosensors were dipped into buffer with or without M2 (the first arrow), followed by immersion into the buffer containing either CD28 or CTLA4 (the second arrow). hB7.1/2-buffer-hB7.1/2 were included as controls to indicate the unoccupied Fc-binding sites on the biosensor. The BLI traces are representative of two independent experiments. Source data are provided as a Source data file.

M2 appears to have a similar immunosuppression function as CPXV M2.

Most viruses have developed extensive immune evasion strategies to facilitate viral replication and transmission, including viral decoy receptors using diverse mechanisms to sequester host immune responses. Most of these decoys work as molecular traps to competitively interact with the relevant ligands and thus block their binding to the natural receptor and antagonize the correlated signaling system[36,37]. Three major classes of viral decoy molecules have been reported based on their function and sequence similarity: (i) viral homologs of host cytokines, chemokines, and interferons, (ii) viral homologs of soluble or inactive versions of signaling receptors, and (iii) viral decoys derived/distinct from host counterparts with novel functions. In this regard, M2, combined with other members of the PIE family, is classified into group iii, which lacks sequence similarity to the host genes but works as a competitive inhibitor for CD28/CTLA4-B7.1/2 signaling.

Among 8 PIE domain family proteins with known structures, most of them are monomeric[22,38–42], and only two of them (Orf virus GIF and CKBP) display as a dimer and bind to their respective ligands with 2:2 stoichiometry[43,44]. Unexpectedly, the cryo-EM analysis shows that M2 displays as hexamers and heptamers and binds to hB7.1/2 with both 6:6 and 7:7 stoichiometries. We found that the oligomeric M2 proteins can engage into hB7.1/2 with higher avidities than the monovalent interactions of M2 with hB7.1/2. In addition, although the monomeric M2 variant is able to bind to hB7.1/2 with sub-micromolar affinities, monomeric M2 cannot significantly inhibit T cell activation mediated by B7.1/2 and CD3, which indicates that multivalent binding effects of M2 to hB7.1/2 play an important role in the subverting the T cell immune response. Similar multivalence binding of viral secreted decoy receptor to its ligand was also reported for Epstein-Barr virus (EBV) encoded BARF1 protein, which forms hexamers and engages to colony-

stimulating factor 1 (CSF-1) with ultrahigh affinity and subverts CSF-1 function effectively[37].

B7.1 and B7.2 play a central role in adaptive immune responses by binding to the costimulatory receptors of CD28 and CTLA4[4,45,46]. Generally, CD28 is believed as a stimulatory molecule and provides a positive second signal for T-cell activation. The B7s-CD28 pathway is also involved in follicular helper T cell function, germinal center formation, and finally affects the humoral immune response[47,48]. In contrast, CTLA4 is thought to mediate the inhibitory pathway and transmit a negative second signal for T cell tolerance and exhaustion[49,50]. CD28 is mainly expressed constitutively on the surface of naïve and activated T cells, whereas CTLA4 is predominantly expressed constitutively on Treg cells and induced on activated T cells upon T cell receptor ligation[51]. CTLA4 has a relatively higher binding affinity to B7 molecules than CD28[52]. CTLA4 may reduce CD28-mediated T cell responses by competition binding with B7.1/2, and previous studies also indicate CTLA4 can affect the CD28 pathway via down-regulation of B7.1 and B7.2 expression on the surface of antigen-presenting cells[53,54]. Our structural data clearly show that the footprints of M2 on human B7 ligands are greatly overlapped with the binding residues of CD28 and CTLA4 on B7 molecules. Furthermore, the cell-based flow cytometry and biochemical data show that the binding of MPXV M2 to B7 ligands drastically sabotages the interactions of B7 molecules with CD28 and CTLA4. Although MPXV M2 may affect both CD28- and CTLA4-mediated T cell activities via shared B7 molecules, we argue that MPXV M2 is likely to help the virus to antagonize host immune response due to the distinct expression patterns and levels of CD28 and CTLA4, especially at the early stage of viral infection when naïve T cell need to be primed to fight against invading pathogens.

In summary, our study demonstrates that MPXV M2 can bind hB7.1/2 and suppress T cell activation mediated by hB7.1/2

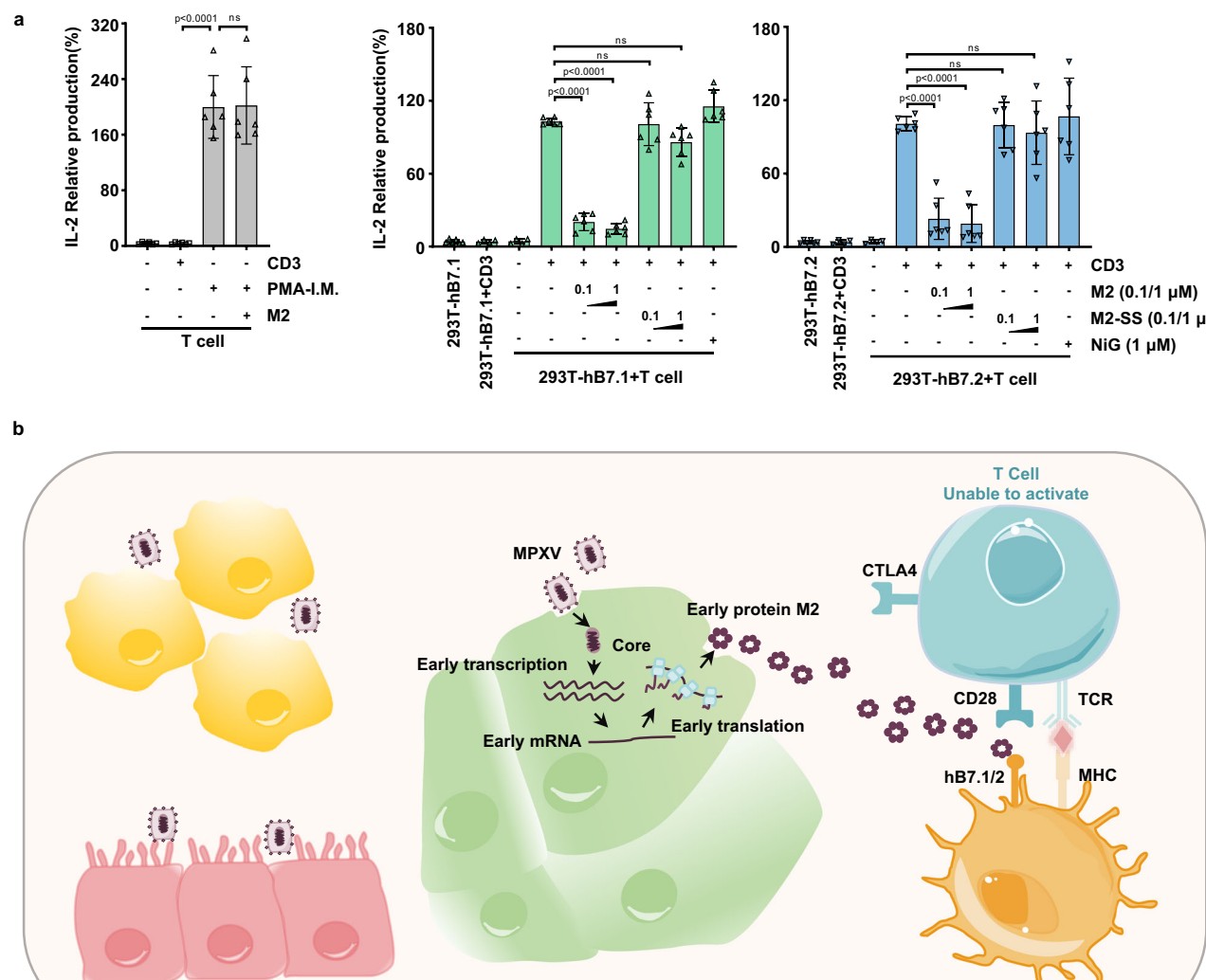

**Fig. 6 | M2 blocks T cell activation mediated by hB7.1/2. a** Purified human T cells were co-cultured with 293T-hB7.1/2 cells on anti-human CD3 pre-coated plates or stimulated with PMA/Ionomycin. The cells were cultured in the medium supplemented with wild-type MPXV M2 (0, 0.1, or 1 μM), or monomeric M2-SS (0, 0.1, or 1 μM) or 1 μM Nipah G (NiG) protein as indicated, and the IL-2 production was analyzed by ELISA at 48 h post stimulation. The data averaged from three independent experiments are shown as means with SD. Data were analyzed by one-way ANOVA with Tukey's multiple comparisons test. ****$P < 0.0001$; ns, non-significance. The T cell sorting strategy is shown in Supplementary Fig. 11c. Source data are provided as a Source data file. **b** Proposed working model of abrogation of T cell activation by M2.

costimulation via the blockade of CD28 binding to hB7.1/2. The amino acids of M2 within M2-hB7.1/2 interfaces are highly conserved among MPXV clades and other orthopoxviruses, and our results combined with previous studies of CPXV and VACV suggest that the regulation of costimulatory and coinhibitory pathways is a conserved immunomodulatory strategy for orthopoxviruses, although the in vivo effect of MPXV M2 on immune suppression requires further study. Our work establishes a structural framework to understand the role of orthopoxvirus M2 in host immunomodulation.

## Methods

### Cloning, protein expression, and purification

The genes of MPXV M2 (NP_536453.1, Clade I), CPXV M2 (NP_619829.1), ectodomain of human B7.1 (hB7.1, NP_005182.1), human B7.2 (hB7.2, NP_787058.5), and CTLA4 (NM_005214.5) were synthesized by Tsingke Biotechnology Co. VACV M2 (AAA48004.1) and MPXV M2-SS (Mutation residues: C25S, C130S) were obtained by point mutation using MPXV M2 DNA as a template. The genes encoding mouse B7.1 (mB7.1)

and B7.2 (mB7.2) were amplified from the cDNA of C57BL/10 mouse spleen cells. The coding sequences of human CD28, PD-L1, PD-L2, B7-H3, B7-H4, B7-H5, B7-H6, B7-H7, and ILDR2 were amplified from the cDNA of Jurkat, THP-1 and A549 cells. The above sequences encoding the extracellular domain were cloned into a mammalian expression vector with a C-terminal 6xHis-tag or human Fc tag, and all plasmids were verified by DNA sequencing.

The vectors were transiently transfected into Expi293 cells with Polyethylenimine (PEI, Polyscience), and the supernatant of cultured cells was collected 108 h after transfection for purification. His-tagged M2 (residues: 18–220), MPXV M2-SS, hB7.1 (residues: 35–234), hB7.2 (residues: 26–238), mB7.1 (residues: 38–245), and mB7.2 (residues: 24–245) were purified by Ni-Charged Resin (GenScript), and hFc-tagged hCD28 (residues: 19–152), hCTLA4 (residues: 36–160), hB7.1, hB7.2, mB7.1, and mB7.2 were purified by protein A beads (Smart-lifesciences). The soluble proteins were further purified by size exclusion chromatography using Superdex 200 Increase 10/300 GL column (Cytiva). In addition, the MPXV M2 gene was fused with an

Avi-tag at the N-terminal and 6xHis-tag at the C-terminal, and then the vector was co-transfected into Expi293 cells with the plasmid expressing biotin ligase (BirA) for the expressing of site-specific biotinylated MPXV M2 (Bio-M2). VACV Bio-M2 and CPXV Bio-M2 were randomly biotinylated with EZ-Link NHS-PEG4-Biotin (Thermo Scientific™) for flow cytometry analysis. The full-length coding sequences of hB7.1 and hB7.2 were subcloned into modified lentiviral transfer vectors pCDH-RFP, which expresses RFP under the IRES sequence as a marker for gene transduction.

## hB7.1/2-transduced cells

Lentiviral transfer vectors containing hB7.1/2 (pCDH-RFP-hB7.1 and pCDH-RFP-hB7.2) were co-transfected with packaging plasmids pMD2G (Addgene, 12259) and psPAX2 (Addgene, 12260) into 293T cells via PEI transfection reagent. Forty-eight hours post-transfection, cell supernatant containing lentivirus was collected and concentrated by Lenti-X Concentrator (Takara). Then 293T cells were transduced with the above lentiviruses carrying hB7.1 or hB7.2 to make transduced cells.

## Biolayer interferometry assay

The binding affinity of purified Bio-M2 protein to human and mouse B7.1/2 was detected by Octet-Red96 BLI. Briefly, Bio-M2 (10 μg/mL) was loaded onto streptavidin biosensors for 3 min in running buffer (10 mM HEPES pH 7.4, 150 mM NaCl, 3 mM EDTA, and 1% BSA), then the M2-loaded sensors were washed with running buffer for 10 s to get rid of non-specific loading proteins. Subsequently, the M2-loaded tips were dipped into the hB7.1/2-containing buffer for 3 min, followed by immersion into the running buffer to record the association and dissociation kinetics.

Similarly, we tested the binding affinity of purified hB7.1/2-hFc to MPXV M2 and M2-SS, mB7.1/2-hFc to MPXV and CPXV M2s with protein A sensors. Specifically, hB7.1/2-hFc and mB7.1/2-hFc were loaded (10 μg/mL) onto protein A biosensors in a running buffer (10 mM HEPES pH 7.4, 150 mM NaCl, 3 mM EDTA, 0.05% Tween, 1% BSA) for 5 or 10 min followed by dipping the sensors into the buffer for 30 s to remove the non-specific loading proteins. Then the h/mB7.1/2-hFc-loaded sensors were immersed in a buffer containing serially-diluted MPXV M2, MPXV M2-SS, or CPXV M2 proteins for the association, followed by immersing the sensors in a running buffer for the dissociation. After each run, the protein A biosensors were dipped into the regeneration solution (10 mM Glycine-HCl pH 2.0) to remove the loaded h/mB7.1/2-hFc proteins. The experiment was conducted at 25 °C, and the binding force and decoupling values were analyzed using Octet data analysis software 12.2.0.20. The sensors without M2 or h/mB7.1/2-hFc loading are used parallelly to define the background.

In the competitive BLI assay, hB7.1/2-hFc proteins were loaded onto the protein A biosensors first, and then the biosensors were dipped into the buffer with/without 5 μM MPXV M2 for 15 min. Subsequently, the biosensors were dipped into the wells containing 5 μM tested proteins (CD28-hFc, CTLA4-hFc, or hB7.1/2-hFc) to monitor the binding signals. At the binding stage, the tested hB7.1/2-hFc proteins were included to define the background binding signal as extra Fc-binding sites may exist on the protein A biosensors. MPXV M2 was considered as a competitor for CD28 and CTLA4 binding to hB7.1/2 as the presence of M2 reduced more than 75% BLI signal compared with the no M2-bound group.

## Cryo-EM sample preparation

MPXV M2 was incubated with respective hB7 at a molar ratio of 1:1.2 on ice for 30 min. An aliquot of 3.5 μL M2 or mixture at 0.8 mg/mL was applied to Au 200 mesh R1.2/1.3 holey carbon grid (Quantifoil). After incubation for 20 s, the grids were blotted for 2 s at 100% humidity and 4 °C, and plunge-frozen in liquid ethane using Vitrobot Mark IV (FEI Thermo Fisher).

## Cryo-EM data collection and image processing

Cryo-EM data acquisition was performed at a CRYO ARM 300 electron microscope (JEOL, Japan) operating at 300 kV with a K3 direct electron detector (Gatan, USA). Cryo-EM images were recorded automatically using Serial-EM software in super-resolution mode with a super-resolution pixel size of 0.475 Å/pixel at defocus values ranging from −0.5 to −2.5 μm at a calibrated magnification of ×50,000. Data were collected at a frame rate of 40 frames per second. The total electron dose was 40 e/Å².

The patch motion correction and CTF estimation of all stacks were performed with cryoSPARCv3.3.1[55]. For the MPXV M2-hB7.1 complex, particles were automatically selected using Blob picker and extracted from the micrographs with a box size of 320 pixels and subsequently subjected to 2D classification. 500,649 good particles were selected for two rounds of ab-initio reconstruction and then heterogeneous refinement. Finally, a predominant class containing 251,261 particles was selected for CTF refinement and non-uniform refinement (NU-refinement), yielding a reconstruction at an overall resolution of 3.04 Å with C6 symmetry.

Data processing for the MPXV M2-hB7.2 complex followed a similar procedure. Briefly, particles were auto-picked by Topaz picker, and 401,493 particles were extracted for 2D classification using a particle box size of 320 pixels. Ab-initio reconstruction and heterogeneous refinement were performed to generate 6 classes using 326,510 particles selected from good 2D classes. One class representing hexamer containing 198,500 particles and one class representing heptamer containing 45,302 particles were selected to perform CTF refinement and NU-refinement with respective C6 and C7 symmetry, resulting in an overall resolution of 2.7 Å hexamer map and an overall resolution of 3.04 Å heptamer map, respectively.

## Model building and refinement

M2 model predicted by AlphaFold2[56] and crystal structures of human B7.1 (PDB: 1DR9) and B7.2 (PDB: 1NCN) were fitted into the cryo-EM maps using Chimera[57]. Cycles of model building and refinement were carried out in Coot[58] and Phenix using phenix.real_space_refine[59] with C6 or C7 symmetry imposed. For human B7.1/2 proteins, only the N-terminal receptor binding domains were built. The quality of the model was analyzed with MolProbity in Phenix[60]. Refinement statistics are summarized in Supplementary Table 1.

## Flow cytometry-based binding and competition assays

The ability of M2 to interact with the B7 family ligands on the cell surface was investigated by flow cytometry using Expi293 cells transiently transfected with B7 ligands. Briefly, the mammalian cell expression vectors containing full-length B7 ligand and RFP genes were constructed, and these plasmids were transiently transfected into Expi293 cells. Forty-eight hours post-transfection, the cells were fixed with 4% paraformaldehyde, and then stained with serial dilutions of Bio-M2 followed by Streptavidin-APC (BD Biosciences) secondary antibody at 1:1000 dilution. The Expi293 cells expressing B7 ligands stained with Streptavidin-APC alone were used as a negative control. Cell debris and dead cells are excluded reliably based on forward and side scatter density plots (FSC-A/SSC-A) and cells with B7s expression (RFP+) were selected to analyze M2 binding positive cells (APC).

We used hB7.1-transduced (293T-hB7.1) and hB7.2-transduced (293T-hB7.2) cells to assess the effect of M2 on the binding of CD28 and CTLA4 to hB7.1 and hB7.2. Briefly, the 293T-hB7.1 and 293T-hB7.2 cells were fixed and incubated with 0.5 μM hCD28-hFc or 0.02 μM hCTLA4-hFc in the presence or absence of 1 μM MPXV M2. Changes in fluorescence signals were detected after staining with secondary Alexa Fluor™ 488-conjugated goat anti-human IgG (H + L) (Invitrogen) diluted at 1:1000. The 293T-hB7.1/2 cells stained with Alexa Fluor™ 488 were used as negative controls, and the expression of B7 family members were evaluated by RFP fluorescence density. 10000 events

# Article

were analyzed and cells with B7s expression (RFP+) were selected to analyze CD28 or CTLA4 binding positive cells (Alexa Fluor™ 488). The fluorescence signal changes were analyzed using CytoExpert (Beckman), and the data were processed and plotted using FlowJo and GraphPad Prism software, respectively.

## Western blot

BHK21 cells were seeded in 24-well tissue culture plates at $2 \times 10^5$ cells/well. One day later, the cells were infected with the vTF7-3 virus in 50:1 or 100:1 (V/V) ratios. Forty-eight hours post-infection, the cell supernatant was collected and centrifuged at $3000 \times g$ for 5 min to remove the cell debris. Purified recombinant proteins MPXV M2 and VACV M2 were included as positive controls. The samples were mixed with loading buffer (125 mM Tris-HCl pH 6.8, 5% SDS, 0.125% bromophenol blue, 25% Glycerol) and subjected on SDS-PAGE gel, and then the protein was transferred onto PVDF membranes. The membrane was blocked in Tris buffer pH 7.0 supplemented with 8% non-fat milk and 0.1% Tween 20 (TBS-T) for 3 h, followed by the incubation with 3 μg/mL hB7.1/2-hFc in TBS-T plus 8% milk overnight at 4 °C, and then washed several times with TBS-T. Anti-human Fc-HRP antibody (Abclonal, 1/20000) was used as the secondary antibody. The membrane was incubated with a super-sensitive ECL chemiluminescent substrate (Biosharp) for 2 min and visualized on Tanon-5200 Chemiluminescent Imaging System (Tanon Science & Technology).

## Isolation of PBMCs from human blood samples

The blood samples of healthy volunteers were collected by Renmin Hospital of Wuhan University, China. Peripheral blood mononuclear cells (PBMCs) were isolated with a human lymphocyte separation tube (DAKEWE, 7922021) based on the manufacturer's instructions. The fresh or frozen PBMCs were stained with CD4-BV510 (BD Horizon™, 562970), CD8-BV510 (BioLegend, 344732), and Fixable Viability Stain 780 (FVS780, BD Horizon™, 565388), then live T cells (FVS780⁻BV510⁺) were sorted and used for T cell activation assay. All the volunteers were provided informed written consent forms, and the procedures for human blood sample collection were approved by the Ethics Committee of Wuhan University with an approval number of WHU-LFMD-IRB2023010.

## Enzyme-linked immunosorbent assay (ELISA)

96-well tissue culture plates were pre-coated with 10 μg/mL anti-CD3 antibody (Invitrogen, 16-0037-81) diluted in PBS at 37 °C for 1 h. Then, the mixture of purified human T cells ($1–1.2 \times 10^5$ cells/well) and 293T-hB7.1 or 293T-hB7.2 ($2.0–2.5 \times 10^4$ cells/well) cells were seeded in anti-CD3 antibody-treated 96-well plates. The RPMI-1640 medium (Monad) supplemented with recombinant M2 (0, 0.1, or 1 μM), M2-SS (0, 0.1, or 1 μM), or 1 μM control proteins (Nipah G, expressed in the same cells and purified similarly as M2 and M2-SS) were used to investigate the M2 effect on the T-cell activation. PMA (81 nM)/Ionomycin (1.5 μM) which can bypass MHC/TCR and B7/CD28 costimulatory signals and activate T cells, is included to serve as a positive control. Forty-eight hours later, the supernatant from cultured cells were collected to measure the IL-2 production by ELISA (Invitrogen, 88-7025-88) according to the manufacturer's protocol.

## Size exclusion chromatography coupled with multi-angle light scattering (SEC-MALS)

The SEC liquid chromatography separation was performed on an Agilent 1260 high performance liquid chromatography (HPLC) system (Agilent, DE, USA). A MALS detector (DAWN®) and a refractive index (RI) detector (Wyatt Technology, Santa Barbara, CA, USA) were connected in series to the UV detector on the SEC system. For each sample, 100 μL at 0.7 ~ 3 mg/mL was injected onto the WTC-030S5 column, 7.8 by 300 mm I.D; particle size, 5 μm; pore size, 300 Å, (Wyatt Technology, Santa Barbara, CA), at a flow rate of 0.5 mL/min in 20 mM Tris pH 8.0, 150 mM NaCl. Bovine Serum Albumin (Thermo Fisher Scientific, Waltham, MA) was used to normalize the static light scattering detector. The data were analyzed using the ASTRA software (6.1.2.84).

## Data availability

All data needed to evaluate the conclusions in the paper are present in the paper or the Supplementary information. The cryo-EM maps have been deposited to Electron Microscopy Data Bank with accession codes EMD-35074, EMD-35075, and EMD-35076. Atomic coordinates have been deposited to the Protein Data Bank (PDB) with accession codes 8HXA, 8HXB and 8HXC. The published structures used in this study are available in the PDB under the following accession codes and links: PDB:1DR9, PDB:1NCN, PDB:1I8L, PDB:1I85, PDB:3ONA, PDB:4HKJ, PDB:5D28, PDB:4ZK9, PDB:2FFK, and PDB:2VGA. All other data supporting the findings of this study are available within the article and its Supplementary information files. Source data are provided with this paper.

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

## Acknowledgements

We thank H. Zhang for the critical suggestions, Y. Zhou and S. Li for their help in the sequence analysis, Y. Huang and X. Cui for the blood sample collection. We thank the Center for Instrumental Analysis and Metrology of Wuhan Institute of Virology for providing technical assistance. This work was supported by the National Key Research and Development

Program (2022YFC2604100 to H.Z. and 2022YFC2303300 to Z.D.), National Natural Science Foundation of China (82172268 to H.Z. and 32200123 to Z.D.), Fundamental Research Funds for the Central Universities (2042022kf1044) and Hubei Province health and family planning scientific research project (WJ2023Q007) to H.Z., and CAS Pioneer Hundred Talents Program to Z.D.

## Author contributions

H.Z. and Z.D. conceived and supervised the project. S.Y. performed biochemical preparations and functional assays with the help of F.Y., R.C., D.Z., Y.Z., and X.R.; Y.W. and Z.D. performed cryo-EM experiments and structural determination; F.Y. generated hB7.1/2-transduced 293T cells and performed SEC-MALS assay. Z.D. built the models. S.Y., Z.D., and H.Z. analyzed the data and prepared the manuscript with input from all authors.

## Competing interests

The authors declare no competing interests.
