## [Peer Review File · Nature Communications]

nature portfolio

Peer Review FileReviewer comments, first round:

Reviewer #1 (Remarks to the Author):

The manuscript by Yang et al. reports well-described and performed structural and mechanistic studies on how the M2 protein from the monkeypox virus sequesters human protein targets, identified in the presented work as B7.1 and B7.2, to achieve immunomodulation of the host.

Although it is important to disseminate the reported complexes of monkeypox virus M2 structures with human B7.1 and B7.2 to the research community as soon as possible, it is also important to make sure that these findings are presented in a scientifically sound way.

I herewith provide several remarks towards improving the presentation of these important and timely findings.

(1) The authors determined that the M2-hB7.2 complex is manifested in two distinct assemblies, with 6:6 and 7:7 stoichiometry respectively.

-The authors do not provide any biochemical data about how many of the reconstituted complexes used for structure determination by cryo-EM were produced. For instance, such data could be added to Extended Data Figure 1.

In principle, size-exclusion chromatography in line with Multi-angle laser light scattering (SEC-MALLS) or mass-photometry should be able to provide insights into the presence of the 7:7 assembly in solution.

Given the covalent character of oligomeric M2 (via cysteine disulfides) the presence of heptameric vs hexameric M2 should be resolvable on an appropriately executed SDS-PAGE analyses.

- The authors do not discuss the biological relevance and implications of the complex with 7:7 stoichiometry. Why did the 7:7 not get observed in the case of the M2-hB7.1 complex? Perhaps it is present in solution but did not get evaluated in any of the obtained cryo-EM 2D classes.

(2) Figure 1D and approach towards evaluating the interaction of M2 with human/mouse B7.1 and B7.3 via Biolayer interferometry.

By immobilizing M2, which is either hexameric or heptameric (as per evidence via cryo-EM), presents a substantial technical complication/challenge in that it presents a binding entity with a 6-fold or 7-fold valence. Therefore, avidity is a key issue here and is set to mask the accurate kinetic quantification of the 1:1 binding mode. Reversing, the immobilization strategy could be one way to resolve this.

Furthermore, the nomenclature for the kinetic binding parameters needs to be corrected throughout the manuscript and in the figures.

k_a , k_d , and K_D should be given with the k/K in italics and their respective subscripts as subscripts in normal font (i.e. not in italics). i.e. k_a , k_d , and K_D .

(3) The Discussion could benefit from analogies seen in the sequestration mechanism employed by other viral decoy receptors (see comprehensive review DOI: 10.1038/nri.2016.134) employing n-order of binding of host proteins to achieve immunomodulation. E.g. See the interaction of BAF1 with human CSF-1 (DOI: 10.1038/nsmb.2367).

(4) The manuscript will benefit substantially from a summarizing figure in which the authors position their structural findings for the sequestration of BL7.1. and BL7.2 by M2 and the consequences of these interactions in an immunological context, e.g. with respect to CD28 and CTLA4 immunology.

(5) Figure 5C: The green and blue lines for the sensorgrams are difficult to distinguish in terms of color, plus the choice of red does not help with colorblindness.

Importantly, the BLI traces for the binding of PD-L1 to a preformed M2-hB7.1 complex are confusing. While there is an indication of a new binding event at the second arrow, whatever has bound appears to dissociate immediately. Please comment.

(6) The current title is not quite right in terms of language usage. It will need to be rephrased to recapitulate more accurately the reported findings.
e.g. "Structures of monkeypox virus M2 with B7.1..." is not correct English to convey the message that this is a paper about the complex of the two proteins.

(7) Extended Data Figure 1: specify explicitly in the legend that BME is β -mercaptoethanol.

(8) Figure 5: This will require rearrangement of panels because the legend calls out figure panels in a non-sequential manner.

Reviewer #2 (Remarks to the Author):

The main incremental advance of this work is the experimental demonstration that MPXV can bind human B7 family proteins B7.1 and B7.2 which are expressed in B cells and dendritic cells. The authors convincingly used flow cytometry to demonstrate this interaction in cells and performed quantitative measurements by BLI showing sub-micromolar binding affinities. Structures of complexes between recombinant M2 and both B7 types were reconstructed from single particle cryo-EM image datasets in 6-fold and 7-fold symmetry, and atomic models were fit and refined into these maps. The manuscript provides a detailed and complete list of a large number of model-derived side chain interactions.

B7 proteins are known to interact with immunomodulatory receptors such as CTLA4 and CD28. The M2 protein of orthopoxviruses has multiple functions – however, the authors single out the binding of M2 to B7 family proteins and suggest that this represents a general mechanism in orthopoxviruses to "sabotage T cell activation". While they successfully show the binding of M2 to B7 proteins in cellulo and in vitro using recombinantly expressed protein, and that this interaction prevents binding of CD28 or CTLA4 to B7, the biological significance of this (expected) observation is not at all clear.

The cryo-EM structures – although technically well done – are not helpful in this regard, because this "first" MPXV structure of M2 has high sequence homology with more than a dozen published structures (ED Fig. 6) that allowed accurate prediction by alphafold2, which fit the cryo-EM structures well as an initial model for refining the atomic coordinates. The authors do not try to explain the observed 6- and 7-fold symmetries in these in vitro single particle assemblies, which have not been reported as an arrangement of M2 in native poxvirus particles in their many forms during the viral life cycle. Based on the presented cryo-EM structures, if M2 would be able to form a hexagonal honeycomb-like lattice as several of the structural proteins in orthopoxviruses do, it would lose its ability to interact with B7 proteins. Therefore, the structural mechanism of poxviral M2 function as advertised in the abstract and throughout the manuscript cannot be explained based on these structures alone.

Without in situ data of the assembly of M2 in actual virions – even using vaccinia virus – the far-reaching biological claims about immune regulation based on the demonstrated structures and interactions are not justified.

Reviewer #3 (Remarks to the Author):

This manuscript by Yang & Wang, et al report on the binding activity and structures of the monkeypox virus M2 protein's interaction with B7.1 (CD80) and B7.2 (CD86). The M2 protein, which is highly conserved in the orthopoxvirus genus has multiple functions, including immune evasion. Prior publications have shown that this protein has a role in uncoating and DNA replication, inhibition of NF-kappaB, and binding to human/murine B7.1 (CD80) and B7.2 (CD86) leading to the blockade of T cell activation and proliferation.

This manuscript confirms results of the previously published binding studies that used highly homologous M2 proteins from vaccinia virus and cowpox virus. The new information in this

manuscript is the Cryo-EM structures of the viral protein's interaction with CD28 and CTLA4 as well as the structure of the M2 monomer.

Given the global outbreak of MPXV with Clade IIb virus, it would likely be useful for authors to indicate that their work is with a protein from a Clade I virus. In addition, it might also be useful indicate if there are any amino acid differences between Clade I protein and Clade IIa and IIb virus or state they are identical. Also, during the global outbreak many mutations in the genome has occurred. Has there been any mutations in the Clade IIb M2 protein (and if any are at a critical location based on cryo-EM structures)?

Does any of the structure data help explain the enhanced interaction between B7.1 and PD-L1 after M2 engagement?

Do the authors want to speculate on the differences in binding activity of the mpox protein to human compared to mouse ligands?

Minor points:

Extended figure 6. Methods section indicates that CPXV M2 (NP_619829.1) sequence was used in experiments, but amino acid alignment in Extended figure 6 is Cowpox virus, CPXV, AAM13487.1. Also, in extended figure 6: Variola virus, VARV, ABF23392.1. Is this sequence from variola minor or variola major?

Line 37. Reference mishap. Reference 2 seems to be the wrong reference.

Line 161. Typo VACA. Should be VACV

Response to Reviewers' Comments:

We greatly appreciate the time, efforts, and constructive comments of reviewers for improving our manuscript. During this revision, we conducted more experiments, including mutation protein characterization, SEC-MALS analysis, and ex vivo T cell activation assay, to corroborate our structural finding. We also revised the text and add more mechanistic discussion to improve the presentation.

Reviewer #1:

The manuscript by Yang et al. reports well-described and performed structural and mechanistic studies on how the M2 protein from the monkeypox virus sequesters human protein targets, identified in the presented work as B7.1 and B7.2, to achieve immunomodulation of the host.

Although it is important to disseminate the reported complexes of monkeypox virus M2 structures with human B7.1 and B7.2 to the research community as soon as possible, it is also important to make sure that these findings are presented in a scientifically sound way. I herewith provide several remarks towards improving the presentation of these important and timely findings.

We greatly appreciate the favorable summary and review.

(1) The authors determined that the M2-hB7.2 complex is manifested in two distinct assemblies, with 6:6 and 7:7 stoichiometry respectively.

-The authors do not provide any biochemical data about how any of the reconstituted complexes used for structure determination by cryo-EM were produced. For instance, such data could be added to Extended Data Figure 1.

We initially mixed M2 with hB7.1/2 at a molar ratio of 1:2 to form complexes and further purified the complexes by size-exclusion chromatography (SEC). The SEC profiles of complexes were provided in **new Extended Data Figure 8A**. However, preferred orientation issues were encountered using the purified complexes for the cryo-EM study, precluding high-resolution reconstruction. We finally solved the preferred orientation issues by directly freezing the mixture of M2 and hB7 proteins at a molar ratio of 1:2.

New Extended Data Figure 8A. Size exclusion chromatography (Superose 6 Increase

10/300 GL cytiva) profiles of MPXV M2, M2-hB7.1/2 complexes, and monomeric M2 variant (M2-SS).

In principle, size-exclusion chromatography in line with multi-angle laser light scattering (SEC-MALLS) or mass-photometry should be able to provide insights into the presence of the 7:7 assembly in solution.

As the review suggested, we have performed SEC-MALS experiments to investigate the oligomeric states of M2 and the binding stoichiometry of M2-B7.1/2 complexes.

SEC-MALS results show that the average protein component molecular weight (MW) of M2 is 154.9 kDa \pm 0.557%, corresponding to $\sim 6.12 \times$ M2 monomer (expected monomeric MW: 25.3 kDa). The result is consistent with our cryo-EM results that both hexameric and heptameric species exist for M2, although separated peaks were not observed, possibly due to the limit of SEC-MALS resolution and the minor population of heptameric M2. The M2-hB7.1 complex peaks of 222.6 kDa, 246.1 kDa, or 249.3 kDa were characterized upon the addition of 1-fold, 2-fold, or 8-fold of hB7.1, respectively. The M2-hB7.2 complex peaks of 219.2 kDa, 233.3 kDa, or 238.5 kDa were also observed upon addition of 1-fold, 2-fold, or 8-fold of hB7.2, respectively. The calculated MWs of the complexes are not consistent in each run to determine the exact stoichiometry, likely owing to the relatively fast dissociation rate of hB7.1/2 to M2.

	M2+hB7.1			M2+hB7.2		
	1:1	1:2	1:8	1:1	1:2	1:8
M2:hB7.1/hB7.2	1:1	1:2	1:8	1:1	1:2	1:8
Total MW	328.2 kDa \pm 0.299%	377.1 kDa \pm 0.741%	389.2 kDa \pm 0.358%	289.8 kDa \pm 0.242%	326.7 kDa \pm 0.738%	349.6 kDa \pm 0.473%
The protein component	222.6 kDa \pm 0.300%	246.1 kDa \pm 0.739%	249.3 kDa \pm 0.357%	219.2 kDa \pm 0.242%	233.3 kDa \pm 0.737%	238.5 kDa \pm 0.469%
The glycan component	105.6 kDa \pm 0.976%	131.0 kDa \pm 2.258%	139.9 kDa \pm 1.059%	70.6 kDa \pm 1.022%	93.4 kDa \pm 2.688%	110.0 kDa \pm 1.565%

New Extended Data Fig. 8C-H. SEC-MALS analysis of wild type M2 (M2 WT), monomeric M2 (M2-SS), hB7.1, hB7.2, M2-hB7.1 and M2-hB7.2 complexes.

In addition, we also collected cryo-EM data of M2 alone. Although we did not obtain

high-resolution reconstruction, partially owing to the severe preferred orientation issue of the sample, the reference-free 2D classification clearly shows that both hexameric and heptameric M2 exist. Combined SEC-MALS data with cryo-EM data of M2-hB7.1 and M2-hB7.2 complexes reveal that M2 can form both hexamer and heptamer, and hB7.1/2 can interact with M2 in 6:6 and 7:7 stoichiometries.

New Extended Data Fig. 8B. 2D class averages of MPXV M2 alone.

Given the covalent character of oligomeric M2 (via cysteine disulfides) the presence of heptameric vs hexameric M2 should be resolvable on an appropriately executed SDS-PAGE analyses.

We apologize for the lack of clarity. For the recombinant M2 from MPXV, VACV and CPXV, we did observe two bands with molecular weights between 150 and 250 kDa in the absence of reduced reagent 2-Mercaptoethanol (BME) by SDS-PAGE analysis, whereas only one single band with a molecular weight of about 35 kDa appeared under the reduced condition in our previous submission (**Extended Data Figure 1A, 1C and 1D**). In addition, we now present new data to show that C25S/C130S (M2-SS) double-cysteine mutant which disrupts inter-M2 disulfide bridge, exists as monomeric state in solution demonstrated by size-exclusion chromatography and SDS-PAGE (**New Extended Data Figure 7 & 8**). As shown in new Extended Data Figures 7 & 8, oligomeric M2 was eluted much faster than the monomeric M2-SS mutant in SEC and there is only one protein band (~35 kDa) was observed for M2-SS with or without 2-Mercaptoethanol on SDS-PAGE. Furthermore, SEC-MALS analysis shows that M2-SS is a monomer (**New Extended Data 8D**). These data were discussed in the main text on page 9 of the Results section (Lines 187-202).

- The authors do not discuss the biological relevance and implications of the complex with 7:7 stoichiometry.

In the revised manuscript we provided new BLI binding data (**Figure 1D-E**) and T-cell activation (**Figure 6A**) data to clearly show that oligomeric M2 uses high avidity to increase the binding affinity and block T-cell activation more efficiently. As the reviewer suggested, we have also expanded the discussion of the biological relevance and implications of the complex with 6:6 and 7:7 stoichiometry on page 15 (lines 319-330).

Why did the 7:7 not get observed in the case of the M2-hB7.1 complex? Perhaps it is present in solution but did not get evaluated in any of the obtained cryo-EM 2D classes. We apologize for the lack of clarity. Two distinct assemblies of 6:6 and 7:7 for both M2-hB7.1 and M2-hB7.2 complexes were observed in cryo-EM. For M2-hB7.1, the majority of particles were in the 6:6 stoichiometry, with a minor proportion found in a 7:7 assembly. It is possible that the 7:7 assembly particles unfavored the freezing condition and did not enter the grid efficiently. In the main text, we stated “*We could not obtain a three-dimensional reconstruction of the M2-hB7.1 heptamer complex, partially due to the few cryo-EM particles of this complex on the cryo-EM grids*”, and we boxed out the 7:7 assembly class in the **Extended Figure 4A**.

(2) Figure 1D and approach towards evaluating the interaction of M2 with human/mouse B7.1 and B7.3 via Biolayer interferometry.

By immobilizing M2, which is either hexameric or heptameric (as per evidence via cryo-EM), presents a substantial technical complication/challenge in that it presents a binding entity with a 6-fold or 7-fold valence. Therefore, avidity is a key issue here and is set to mask the accurate kinetic quantification of the 1:1 binding mode. Reversing, the immobilization strategy could be one way to resolve this.

Thank the reviewer for this great suggestion. As the reviewer suggested, we now have immobilized the hB7.1/2 on the sensors and evaluated the binding properties of M2 to hB7.1/2. In this setting, immobilized sensors are dipped into serial dilutions of M2, which is a mixture of hexamer and heptamer in the solution and is supposed to interact with hB7.1/2 in a more biological condition. As shown in the **new Figure 1**, M2 could bind to hB7.1/2 with significantly higher affinity and slower dissociation rate compared to the reversing binding scenario due to its multivalence binding properties. In addition, we have generated M2 variant, which lost the intermolecular disulfide linkage and formed a monomer in solution, and the interaction of monomeric M2 variant with immobilized B7.1/2 were also assessed. We have added this data in the **new Extended Data Figure 7** and discussed these data on page 5-6 and 10 of the Results section.

New Figure 1D-E. Quantitative analysis of the binding affinity of MPXV M2 to hB7.1/2 by BLI.

New Extended Data Figure 7. Binding of recombinant monomeric M2 variant (M2-SS) protein to hB7.1 and hB7.2.

Furthermore, the nomenclature for the kinetic binding parameters needs to be corrected throughout the manuscript and in the figures.

k_a , k_d , and K_D should be given with the k/K in italics and their respective subscripts as subscripts in normal font (i.e. not in italics). i.e. k_a , k_d , and K_D .

We thank the reviewer for the point. We have corrected this nomenclature of the kinetic binding parameters throughout the manuscript.

(3) The Discussion could benefit from analogies seen in the sequestration mechanism employed by other viral decoy receptors (see comprehensive review DOI: 10.1038/nri.2016.134) employing n-order of binding of host proteins to achieve immunomodulation. E.g. See the interaction of BAF1 with human CSF-1 (DOI: 10.1038/nsmb.2367).

We have now included discussion about high avidity binding feature of oligomeric M2 in Discussion (p. 15).

(4) The manuscript will benefit substantially from a summarizing figure in which the authors position their structural findings for the sequestration of BL7.1. and BL7.2 by M2 and the consequences of these interactions in an immunological context, e.g. with respect to CD28 and CTLA4 immunology.

This is a good point and we now present new **Figure 6B** to summarize the discovery.

(5) Figure 5C: The green and blue lines for the sensorgrams are difficult to distinguish in terms of color, plus the choice of red does not help with colorblindness.

Importantly, the BLI traces for the binding of PD-L1 to a preformed M2-hB7.1 complex are confusing. While there is an indication of a new binding event at the second arrow, whatever has bound appears to dissociate immediately. Please comment.

We have used different colors for the BLI traces. While in our previous submission, we used PD-L1 as a positive control to show M2-bound B7.1 is functional, but M2-bound B7.1 lost the binding capacity to CD28 or CTLA4. To clarify this point, we now re-conduct the competition assays with different settings. As shown in new **Figure 5E-H**,

CD28 and hCTLA4 bound to hB7.1/2 very well in the absence of M2, while the pre-bound hB7.1/2 with M2 competitively blocks B7.1/2-CD28 and B7.1/2-CTLA4 interactions.

New Figure 5E-H. CD28/CTLA4 and M2 competition for the binding to hB7.1 and hB7.2 were tested by BLI.

(6) The current title is not quite right in terms of language usage. It will need to be rephrased to recapitulate more accurately the reported findings.

e.g. “Structures of monkeypox virus M2 with B7.1...” is not correct English to convey the message that this is a paper about the complex of the two proteins.

As the reviewer suggested, we have edited the title into “*Structures of monkeypox virus M2 complexed with human B7.1 and B7.2 reveal insight into poxvirus modulation of T cell costimulation*”.

(7) Extended Data Figure 1: specify explicitly in the legend that BME is β -mercaptoethanol.

We have indicated “BME” as “2-Mercaptoethanol”.

(8) Figure 5: This will require rearrangement of panels because the legend calls out figure panels in a non-sequential manner.

We have re-arranged Figure 5.

Reviewer #2 (Remarks to the Author):

The main incremental advance of this work is the experimental demonstration that MPXV can bind human B7 family proteins B7.1 and B7.2 which are expressed in B cells and dendritic cells. The authors convincingly used flow cytometry to demonstrate this interaction in cells and performed quantitative measurements by BLI showing sub-micromolar binding affinities. Structures of complexes between recombinant M2 and both B7 types were reconstructed from single particle cryo-EM image datasets in 6-fold and 7-fold symmetry, and atomic models were fit and refined into these maps. The manuscript provides a detailed and complete list of a large number of model-derived side chain interactions.

B7 proteins are known to interact with immunomodulatory receptors such as CTLA4 and CD28. The M2 protein of orthopoxviruses has multiple functions – however, the authors single out the binding of M2 to B7 family proteins and suggest that this

represents a general mechanism in orthopoxviruses to “sabotage T cell activation”. While they successfully show the binding of M2 to B7 proteins in cellulo and in vitro using recombinantly expressed protein, and that this interaction prevents binding of CD28 or CTLA4 to B7, the biological significance of this (expected) observation is not at all clear.

To address the reviewer’s concern, we have added new experimental data evaluating the biological impact of MPXV M2 on T cell activation. As shown in new Figure 6, addition of recombinant oligomeric M2 significantly suppressed hB7.1/2-mediated T cell activation measured by cytokine IL-2 production. Noteworthy, no inhibition effects were observed for PMA/Ionomycin-activated T cells in the presence of oligomeric M2, and the subversion of the T-cell activation mediated by B7 ligands was significantly reduced for the monomeric M2 variant (new Figure 6A). These data reveal that oligomeric MPXV M2 is able to interact with B7.1/2 and consequently subvert T cell immune response, and multivalence binding features of oligomeric M2 to hB7.1/2 is important for viral evasion of T cell response. The data are now discussed on page 13 of the Results section (lines 267-284).

New Figure 6A: M2 blocks T cell activation mediated by hB7.1/2.

The cryo-EM structures – although technically well done – are not helpful in this regard, because this “first” MPXV structure of M2 has high sequence homology with more than a dozen published structures (ED Fig. 6) that allowed accurate prediction by alphafold2, which fit the cryo-EM structures well as an initial model for refining the atomic coordinates.

We agree with the reviewer that the alphafold2 is a powerful tool and able to predict structures of some proteins with high accuracy; however, the prediction of the protein complex, oligomeric state of protein, and precise interaction interface remains a challenge.

We apologize for not presenting the data clearly. Actually, M2 shares low sequence identity to available structures of the poxviral immune evasion (PIE) family proteins (sequence identity: 14% with ectromelia virus CrmD C-terminal SECRET domain, 17% with cowpox virus CPXV203, 12% with Orf virus GIF, 17% with Orf virus CKBP, 15% with rabbitpox virus vCCI, and 14% with vaccinia virus A41). Although the core domains of the PIE family proteins share generally conserved β -sandwich folding as

indicated in the Extended Data Figure 10, the unique decorations around the core domain usually confer ligand specificities and engage the interface. Our complex structures of M2 with B7 ligands show that M2 exists as a hexamer and a heptamer, and B7 primarily binds to the $\beta 8$ and $\beta 11$ strands. While most of reported PIE family proteins are monomeric state, $\beta 8$ is absent in Orf GIF, CKBP, rabbitpox vCCI and vaccinia A41, and $\beta 11$ is lacking in CPXV203, which further suggests that high-resolution cryo-EM structures are informative for the protein-protein interaction and oligomeric state analysis.

The authors do not try to explain the observed 6- and 7-fold symmetries in these in vitro single particle assemblies, which have not been reported as an arrangement of M2 in native poxvirus particles in their many forms during the viral life cycle. Based on the presented cryo-EM structures, if M2 would be able to form a hexagonal honeycomb-like lattice as several of the structural proteins in orthopoxviruses do, it would lose its ability to interact with B7 proteins. Therefore, the structural mechanism of poxviral M2 function as advertised in the abstract and throughout the manuscript cannot be explained based on these structures alone.

We appreciate the reviewer's comments. Previous study has shown that VACV M2 is secreted as a homo-oligomer by infected cells, and the culture supernatants from several related orthopoxvirus-infected cells can interfere with the binding of CTLA4 and CD28 to B7.1/2 (Kleinpeter et al., 2019). Another study has also shown that the supernatants from wild-type CPXV can potently suppress B7-mediated T cell activation compared to M2-deficient CPXV. In addition, M2-deleted CPXV induced a stronger cellular immune response in mice than WT CPXV (Wang et al., 2019).

In our investigations, recombinant MPXV M2 is able to interact with both B7.1/2 displayed on the cell surface and recombinant B7.1/2 protein in solution. In addition, we have performed a new ex vivo T cell activation assay and found that oligomeric M2 can sharply inhibit T cell activation induced by CD3 antibody and B7.1/2, but the inhibition effects were significantly reduced for the monomeric M2 variant (**new Figure 6A**), these data further demonstrate that poxviral M2 is able to interact with B7.1/2 and consequently subvert T cell immune response. These data are now discussed on page 13.

Without in situ data of the assembly of M2 in actual virions – even using vaccinia virus – the far-reaching biological claims about immune regulation based on the demonstrated structures and interactions are not justified.

As the reviewer suggested, we assessed the M2 assembly in culture supernatant from vaccinia virus (strain vTF7-3) infected cells via western blot, where high molecular weight protein bands were probed by hB7.1/2 under non-reduced conditions (New **Extended Data Figure 9**). The oligomeric feature of M2 has also been demonstrated by a previous study which showed that high molecular weight protein secreted from vaccinia virus (Copenhagen) infected cells can be recognized by hB7.1/2, further liquid chromatography coupled to tandem mass spectrometry (LC-MS/MS) analysis revealed that the hB7.2-binding molecule is VACV M2 protein (Kleinpeter et al., 2019). These

data are now discussed on page 10 of the Results section (lines 203-216).

New Extended Data Figure 9. hB7.1/2 interact with secreted M2 proteins from vaccinia virus-infected cells.

References:

17. Wang, X. et al. Cowpox virus encodes a protein that binds B7.1 and B7.2 and subverts T cell costimulation. *Proc. Natl. Acad. Sci. U. S. A.* 116, 21113–21119 (2019).
18. Kleinpeter P, Remy-Ziller C, Winter E, Gantzer M, Nourtier V, Kempf J, Hortelano J, Schmitt D, Schultz H, Geist M, Brua C, Hoffmann C, Schlesinger Y, Villeval D, Thioudellet C, Erbs P, Foloppe J, Silvestre N, Fend L, Quemeneur E, M. J. By Binding CD80 and CD86, the Vaccinia Virus M2 Protein Blocks Their Interactions with both CD28 and CTLA4 and Potentiates CD80 Binding to PD-L1. *J. Virol.* 93, e00207-19 (2019).

Reviewer #3 (Remarks to the Author):

This manuscript by Yang & Wang, et al report on the binding activity and structures of the monkeypox virus M2 protein's interaction with B7.1 (CD80) and B7.2 (CD86). The M2 protein, which is highly conserved in the orthopoxvirus genus has multiple functions, including immune evasion. Prior publications have shown that this protein has a role in uncoating and DNA replication, inhibition of NF-kappaB, and binding to human/murine B7.1 (CD80) and B7.2 (CD86) leading to the blockade of T cell activation and proliferation.

This manuscript confirms results of the previously published binding studies that used highly homologous M2 proteins from vaccinia virus and cowpox virus. The new information in this manuscript is the Cryo-EM structures of the viral protein's interaction with CD28 and CTLA4 as well as the structure of the M2 monomer.

We appreciate the summary and review.

Given the global outbreak of MPXV with Clade IIb virus, it would likely be useful for authors to indicate that their work is with a protein from a Clade I virus. In addition, it might also be useful indicate if there are any amino acid differences between Clade I

protein and Clade IIa and IIb virus or state they are identical. Also, during the global outbreak many mutations in the genome has occurred. Has there been any mutations in the Clade IIb M2 protein (and if any are at a critical location based on cryo-EM structures)?

We agree with the reviewer that it would be helpful to indicate the M2 sequence information and accordingly have mentioned this in the method (p. 17) and discussion (p. 16). As the reviewer suggested, we compared M2 proteins among Clade I, Clade IIa, and IIb MPXV and found that the M2 amino acids are strictly conserved within available Clade I strains (n=47). Two sequences contain different amino acids in positions 67, 77, and 136, but none of these amino acids are involved in the interaction of M2 with B7.1/2 ligands. In comparing our M2 with that of Clade IIa, P27H and I78V mutations were found in Clade IIa; however, these two residues didn't reside in the interface of the M2-B7.1/2 complexes. Several mutations were identified on M2 cross clade IIb (n=4823) with extremely low frequency (<1.95%); among these mutations, only three sequences carrying P140S mutation slightly contribute to non-bonded contacts in the M2-hB7.2 complex, but not in the M2-hB7.1 complex.

Further, sequence comparison of M2 from different orthopoxviruses reveals that the key contacts in the complexes on the MPXV M2 protein are strictly conserved among distinct representative orthopoxviruses (**Extended Data Figure 10**), suggesting a commonly used T-cell immunomodulatory function and mechanism utilized by poxviral M2. These data are now discussed on page 16.

Does any of the structure data help explain the enhanced interaction between B7.1 and PD-L1 after M2 engagement?

We appreciate the reviewer's point. While previous studies indicated that engagement of CPXV and VACV M2 to B7.1 could increase the interaction of PD-L1 to B7.1, we didn't observe this enhancement effect in our experiments via flow cytometry. The complex structure of the B7.1 variant (ALPN-202) with PD-L1 is available now (PDB: 7TPS), which reveals that PD-L1 recognizes the dimer interface of the hB7.1. The M2-B7.1 complex was superimposed on the PD-L1-ALPN-202 complex using B7.1 as a reference, and we found that the overall B7.1 structure is similar and no significant change was observed with a root-mean-square deviation (rmsd) of 0.84 Å over 106 Ca atoms. As expected, M2 and PD-L1 have distinct binding footprints on B7.1, consistent with our competition assay that the binding of M2 to B7.1 is unable to block the binding of PD-L1 to B7.1. However, the enhancement of binding between B7.1 and PD-L1 after M2 engagement awaits further characterization.

In our previous submission, we used PD-L1 as a positive control to show that M2-bound B7.1 is functional and still able to interact with PD-L1, but M2-bound B7.1 lost the binding capacity to CD28 or CTLA4. To clarify this point, we now re-conduct the competition assays with different settings and delete the PD-L1 data from our revised manuscript (**new Figure 5**).

Do the authors want to speculate on the differences in binding activity of the mpox protein to human compared to mouse ligands?

Since MPXV infects both humans and mice, it is unsurprising that MPXV M2 binds to both human and mouse B7.1/2. We found that MPXV M2 could bind to mouse B7.1 with high avidity, while it shows a relatively weak binding capacity to mouse B7.2. The variation of contact residues may contribute to these differences. However, without the high-resolution structures of MPXV M2-mB7.1/2, we feel it is a challenge and premature to explain these only based on sequence alignment result.

Minor points:

Extended figure 6. Methods section indicates that CPXV M2 (NP_619829.1) sequence was used in experiments, but amino acid alignment in Extended figure 6 is Cowpox virus, CPXV, AAM13487.1. Also, in extended figure 6: Variola virus, VARV, ABF23392.1. Is this sequence from variola minor or variola major?

Many thanks for the reviewer's reminder. The amino acids of M2 between NP_619829.1 and AAM13487.1 are identical, and we have updated the strain information in the Extended Figure 6 legend. The variola virus (ABF23392.1) used for the sequence comparison is variola major, and we have also indicated this in the figure legend.

Line 37. Reference mishap. Reference 2 seems to be the wrong reference.
Thanks. We have fixed this reference.

161. Typo VACA. Should be VACV
We have changed "VACA" to "VACV".

Reviewer comments, further round:

Reviewer #1 (Remarks to the Author):

The authors have responded well to my input and should be commended for how thoroughly they have addressed the initial issues raised and for introducing so much new data and analyses.

Reviewer #3 (Remarks to the Author):

This is a revised manuscript that the authors successfully respond to reviewers' comments.

Only minor comments:

Lines 20-206. Would suggest inserting MPXV into the following sentence and also include reference for vTF7-3.

We here used vaccinia virus (vTF7-3 [ref]) as an indicator for the oligomeric feature of MPXV M2 mainly due to the biosafety concern of working with MPXV and the high conservations ..."

Reference for vTF7-3:

Fuerst TR, Niles EG, Studier FW, Moss B. Eukaryotic transient-expression system based on recombinant vaccinia virus that synthesizes bacteriophage T7 RNA polymerase. Proc Natl Acad Sci U S A. 1986 Nov;83(21):8122-6. doi:10.1073/pnas.83.21.8122. PMID: 3095828; PMCID: PMC386879.

Line 353. Would not refer to three clades of MPXV since there are technically two clades with Clade 2 subdivided. So would just drop three and have the sentence read, "The amino acids of M2 within M2-hB7.1/2 interfaces are highly conserved among the MPXV clades and other orthopoxviruses ..."

Reviewer #4 (Remarks to the Author):

The authors have addressed all reviewers concerns (1) by performing additional experiments on evaluating B7.1/2-mediated T cell activity upon introducing MPXV M2 protein in its oligomeric form; (2) by presenting novelty of the oligomeric Cryo-EM structure of M2 in comparison to other poxviral immune evasion proteins that have low sequence homology; (3) by providing appropriate references and experimental evidences that suggest importance of oligomerization for M2 protein's immunomodulatory effects; and (4) by performing experiments to confirm their findings (M2 oligomerization and interaction B7.1/2) in context of actual virions (vaccinia virus).

The manuscript provides great insights into poxvirus immunomodulation mechanism, and will further our understanding in viral infection in general.

REVIEWERS' COMMENTS

Reviewer #1 (Remarks to the Author):

The authors have responded well to my input and should be commended for how thoroughly they have addressed the initial issues raised and for introducing so much new data and analyses.

Thanks for the review and positive feedback.

Reviewer #3 (Remarks to the Author):

This is a revised manuscript that the authors successfully respond to reviewers' comments.

Thanks for the review and positive feedback.

Only minor comments:

Lines 20-206. Would suggest inserting MPXV into the following sentence and also include reference for vTF7-3.

We here used vaccinia virus (vTF7-3 [ref]) as an indicator for the oligomeric feature of MPXV M2 mainly due to the biosafety concern of working with MPXV and the high conservations ...”

Thanks. We have added the reference and rephrased the sentence accordingly.

Reference for vTF7-3:

Fuerst TR, Niles EG, Studier FW, Moss B. Eukaryotic transient-expression system based on recombinant vaccinia virus that synthesizes bacteriophage T7 RNA polymerase. Proc Natl Acad Sci U S A. 1986 Nov;83(21):8122-6. doi:10.1073/pnas.83.21.8122. PMID: 3095828; PMCID: PMC386879.

Line 353. Would not refer to three clades of MPXV since there are technically two clades with Clade 2 subdivided. So would just drop three and have the sentence read, “The amino acids of M2 within M2-hB7.1/2 interfaces are highly conserved among the MPXV clades and other orthopoxviruses ...”

As the reviewer suggested, we have deleted “three” in the sentence.

Reviewer #4 (Remarks to the Author):

The authors have addressed all reviewers concerns (1) by performing additional experiments on evaluating B7.1/2-mediated T cell activity upon introducing MPXV M2 protein in its oligomeric form; (2) by presenting novelty of the oligomeric Cryo-EM structure of M2 in comparison to other poxviral immune evasion proteins that have low

sequence homology; (3) by providing appropriate references and experimental evidences that suggest importance of oligomerization for M2 protein's immunomodulatory effects; and (4) by performing experiments to confirm their findings (M2 oligomerization and interaction B7.1/2) in context of actual virions (vaccinia virus).

The manuscript provides great insights into poxvirus immunomodulation mechanism, and will further our understanding in viral infection in general.

We greatly appreciate the favorable summary and review.